# Lookaround Optimizer: $k$ steps around, 1 step average

**Jiangtao Zhang**[1], **Shunyu Liu**[1], **Jie Song**[1, †], **Tongtian Zhu**[1], **Zhengqi Xu**[1], **Mingli Song**[1, 2]

[1]Zhejiang University, [2]Hangzhou City University

{zhjgtao, liushunyu, sjie, raiden, xuzhengqi, brooksong}@zju.edu.cn

## Abstract

Weight Average (WA) is an active research topic due to its simplicity in ensembling deep networks and the effectiveness in promoting generalization. Existing weight average approaches, however, are often carried out along only one training trajectory in a post-hoc manner (*i.e.*, the weights are averaged after the entire training process is finished), which significantly degrades the diversity between networks and thus impairs the effectiveness. In this paper, inspired by weight average, we propose **Lookaround**, a straightforward yet effective SGD-based optimizer leading to flatter minima with better generalization. Specifically, Lookaround iterates two steps during the whole training period: the *around* step and the *average* step. In each iteration, 1) the *around* step starts from a common point and trains multiple networks simultaneously, each on transformed data by a different data augmentation, and 2) the *average* step averages these trained networks to get the averaged network, which serves as the starting point for the next iteration. The *around* step improves the functionality diversity while the *average* step guarantees the weight locality of these networks during the whole training, which is essential for WA to work. We theoretically explain the superiority of Lookaround by convergence analysis, and make extensive experiments to evaluate Lookaround on popular benchmarks including CIFAR and ImageNet with both CNNs and ViTs, demonstrating clear superiority over state-of-the-arts. Our code is available at https://github.com/Ardcy/Lookaround.

## 1 Introduction

Recent research on the geometry of loss landscapes in deep neural networks has demonstrated *Linear Mode Connectivity* (LMC): two neural networks, if trained similarly on the same data starting from some common initialization, are linearly connected to each other across a path with near-constant loss [33, 9]. LMC reveals that neural network loss minima are not isolated points in the parameter space, but essentially forms a connected manifold [8, 35]. It has recently been attracting increasing attention from research communities, attributed to its great potential into motivating new tools to more efficiently reuse trained networks. A simple but effective strategy is Weight Average (WA), which directly averages the network weights of multiple trained [46] or sampled [19, 3] networks along the training trajectories for flatter minima and thus better generalization.

WA has become a popular approach in various fields for its efficiency in parameters and effectiveness in performance. For example, Izmailov *et al.* [19] show that *Stochastic Weight Averaging* (SWA) of multiple points along the trajectory of SGD, with a cyclical or constant learning rate, leads to better generalization than conventional training. Following that, Cha et al. [3] propose *Stochastic Weight Averaging Densely* (SWAD), *i.e.*, averaging weights for every iteration in a region of low validation loss , to capture flatter minima and thus achieve superior out-of-domain generalizability. Recently, Wortsman *et al.* [46] propose averaging weights of multiple fine-tuned models with various hyperparameters, coined *Model Soups*, to improve accuracy without increasing inference time.

---

†Corresponding author

37th Conference on Neural Information Processing Systems (NeurIPS 2023).

Generally speaking, WA guides the model points around the loss basin into the interior of the loss basin to find a flatter solution, which is shown to approximate ensembling the predictions [19].

Albeit striking results achieved in some cases, WA easily breaks down owing to the *diversity-locality* conflicts. In one end, when WA is carried out within only one optimization trajectory after the training converges (as SWA does), the function diversity of the sampled checkpoints whose weights are averaged is limited, impair-

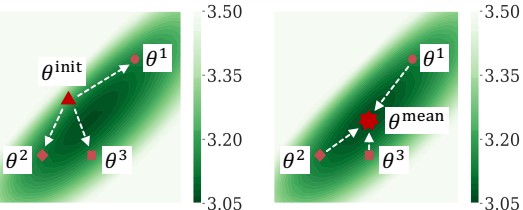

Figure 1: Test set loss landscape. (Left) Around step for diversity. (Right) Average step for locality.

ing the benefits of WA in ensembling. In the other end, if WA is conducted between models trained independently, WA can result in a completely invalid model due to the large barrier between different local minima (violating the locality principle that solutions should be in the same loss basin and closed to each other in parameter space). Prior WA approaches are either limited in function diversity or easily violating the locality principle, hindering its effectiveness and application in practice.

In this work, we propose *Lookaround*, a straightforward yet effective SGD-based optimizer to balance the both. Unlike prior works where WA often carried out along only one training trajectory in a post-hoc manner, Lookaround adopts an iterative weight average strategy throughout the whole training period to constantly make the diversity-locality compromise. Specifically, in each iteration, Lookaround consists of two steps: the *around* step and the *average* step, as shown in Figure 1. The *around* step starts from a common point and trains multiple networks simultaneously, each on transformed data by a different data augmentation, resulting in higher function diversity between trained networks. The *average* step averages these trained networks to get the averaged network, which serves as the starting point for the next iteration, guarantees the weight locality of these networks during the whole training. Lookaround iteratively repeats the two steps constantly, leading the averaged model to a flatter minima and thus enjoying better generalization. We theoretically explain the superiority of Lookaround by expected risk and convergence analysis, and make extensive experiments to evaluate Lookaround on popular benchmarks including CIFAR and ImageNet with both CNNs and ViTs, demonstrating clear superiority over state-of-the-arts.

The contributions of this work can be summarized as follows.

- We propose *Lookaround*, a SGD-based optimizer that enjoys a *diversity-in-locality* exploration strategy. To seek flat minima within the loss basins, Lookaround iteratively averages the trained networks starting from one common checkpoint with various data augmentation.

- We theoretically analyze the convergence of different competitive optimizers in the quadratic case, and prove that our Lookaround show lower limiting expected risk and faster convergence.

- Extensive experiments conducted on the CIFAR and ImageNet benchmarks demonstrate that various CNNs and ViTs equipped with the proposed Lookahead optimizer can yield results superior to the state-of-the-art counterparts, especially the multi-model ensemble methods.

## 2 Related Work

**Mode Connectivity.** Mode connectivity is an intriguing phenomenon in deep learning where independently trained neural networks can have their minima connected by a low-loss path on the energy loss landscape [12, 14]. Some early works [12, 8] find that different independently trained networks can be connected by curves or multiple polylines in low-loss basins. Based on this, some later works [35, 11] show that when training from pretrained weights, the model stays in the same basin of the loss landscape, and different instances of such models can be connected by a straight line. Such findings motivates weight averaging, which averages the weights of different models to obtain higher performance. Recently, weight averaging has been widely used in the fields of neural language processing [21, 32, 39] and multi-objective optimization [33]. We aim to use model connectivity to give our network access to lower positions in the loss basin. In this paper, we propose a simple yet effective method to demonstrate the practical significance of this study.

**Weight Averaging.** While weight averaging shares some similarities with the aggregation operations in Graph Neural Networks [24, 45, 23, 22], its primary application is distinct. In the context of

neural architectures, weight averaging typically represents an overall averaging of weights across different models with the same architecture, diverging from the node-based aggregation in GNNs. Historically, the general idea of weight averaging can trace its roots back to convex optimization [41, 38]. And with the breakthrough of loss landscape visualization technology [12, 28], many applications have applied this idea to neural networks [12, 49, 34]. The concept of weight averaging has been integrated into single-trajectory training [12, 19, 49] and has seen extensive application in distributed setups [29, 31, 50]. Specifically, [19] found that the models trained with SGD optimizer often fell on the edge of a flat basin. Based on that, they propose the SWA method, which uses one-time weight averaging of multiple network points after the entire training process is finished, leading the network down to lower locations in the basin. The later work [49] proposes an optimizer that continuously uses weight averaging to update model weight to improve performance and robustness. Averaging weight during training is more effective than the averaging after training. Unlike single-trajectory optimization strategy, the recent Model Soups method [46] is inspired by mode connectivity, which starts from a typical pretrained weight and averages final fine-tuned models to improve generalizability. These methods inspire us to employ weight averaging method with multi-data augmentations to improve convergence and generalizability.

**Ensemble.** The ensemble learning is a traditional technology that combines multiple model outputs to achieve better robustness and performance [6, 1, 2, 16, 27]. Ensemble methods usually need to train different models to reduce variance and improve the prediction effect. However, in recent years, some methods [18, 12] can obtain different model checkpoints to conduct the ensemble in a single trajectory of model training, which significantly reduces the training time. Note that these methods all require separate inference through each model, which adds to the calculation cost. In contrast, our method does not require additional inference calculations.

## 3 Method

In this section, we present our optimization method Lookaround and provide an in-depth analysis of its properties. In Section 3.1, we provide a detailed description of the Lookaround optimizer. In Section 3.2, we analyze the expected risk of Lookaround, then thoroughly study its convergence at various levels and compare it with the existing optimizer Lookahead [49] and SGD [40]. Intuitively, our proposed optimizer seeks flat minima along the training trajectory by iteratively looking for the diverse model candidates around the trained points, hence called Lookaround. The pseudo-code of Lookaround is provided in Algorithm 1.

### 3.1 Lookaround Optimizer

In this subsection, we introduce the details of Lookaround. In Lookaround, the training process is divided into

---

**Algorithm 1** Lookaround Optimizer.

---

**Require:** Initial parameters $\phi_0$, objective function $\mathcal{L}$, data augmentation list AUG of size $d$, synchronization period $k$, optimizer $A$, dataset $\mathcal{D}$, numbers of training epochs $E$.
  **for** $epoch = 1$ to $E$ **do**
    Synchronize parameters
    **for** $j = 1, 2, \ldots, d$ **do**
      $\theta_{t,j,0} \leftarrow \phi_{t-1}$
    **end for**
    # Around Step: Independent model training.
    **for** $i = 1, 2, \ldots, k$ **do**
      sample minibatch of data $B \sim \mathcal{D}$
      **for** $j = 1, 2, \ldots, d$ **do**
        $\theta_{t,j,i} \leftarrow \theta_{t,j,i-1} + A(\mathcal{L}, \theta_{t,j,i-1}, \mathrm{AUG}_j(B))$
      **end for**
    **end for**
    # Average Step: Weight averaging.
    Compute average weight $\theta_{t,*,k} \leftarrow \frac{1}{d} \sum_{j=1}^{d} \theta_{t,j,k}$
    Perform update $\phi_t \leftarrow \theta_{t,*,k}$
  **end for**
  **return** parameters $\phi$

---

multiple intervals of length $k$, with each interval consisting of an "around step" and an "average step". Let $\phi_0$ denote the initial weights. At $t^{th}$ round of Lookaround, the weights are updated from $\phi_{t-1}$ to $\phi_t$ according to the around step and the average step. The resulting weights $\phi_t$ then serve as the starting point for the subsequent round.

**Around Step.** Given the optimizer $A$, the objective function $\mathcal{L}$, and the stochastic mini-batch of data $B$ from train dataset $\mathcal{D}$. In the around step, $d$ different models are independently trained under $d$

different data augmentations $\text{AUG} = \{\text{AUG}_1, \ldots, \text{AUG}_d\}$ for $k$ batch steps, as follows:

$$\theta_{t,j,k} = \theta_{t,j,k-1} + A(\mathcal{L}, \theta_{t,j,k-1}, \text{AUG}_j(B)). \tag{1}$$

Thanks to the data augmentations, each model is trained in period $k$ to a diverse location in the loss landscape scattered around $\phi_{t-1}$. The Around Step allows optimizing the region surrounding each model iterate, which helps the search of flatter minima. Moreover, note that as the $d$ different models are trained independently, we can use parallel computing to speed up the whole training process.

**Average Step.** In the average step, we simply average $\theta_{t,i,k}$, the weights of each independently trained model to obtain $\phi_t$ for the next around step as follows:

$$\phi_t = \frac{1}{d} \sum_{i=1}^{d} \theta_{t,i,k}. \tag{2}$$

It has been observed that models with the same initialization and trained on different data augmentations exhibit similar loss basins [35], with the models scattered around the edges of these basins. Therefore, incorporating a simple averaging technique into the training process may facilitate the convergence of the models towards lower-loss regions. In Appendix A, we demonstrate how this technique effectively guides the model to the interior of the loss basin.

## 3.2 Theoretical Analysis

### 3.2.1 Noisy Quadratic Analysis

The quadratic noise function has been commonly adopted as an effective base model to analyze optimization algorithms [42, 49, 47, 25], where the noise incorporates stochasticity introduced by mini-batch sampling. In this subsection, we analyze the steady-state risk of our proposed optimizer on the quadratic noise function to gain insights into the performance of Lookaround, and compare the result with those obtained using SGD and Lookahead.

The quadratic noise model is defined as $\hat{\mathcal{L}}(\theta) = \frac{1}{2}(\theta - \mathbf{c})^T \mathbf{A}(\theta - \mathbf{c})$. Following Zhang et al. [49], we assume that the noise vector $\mathbf{c} \sim \mathcal{N}(\theta^*, \Sigma)$, both $\mathbf{A}$ and $\Sigma$ are diagonal matrices, and that the optimal solution $\theta^* = \mathbf{0}$. We denote $a_i$ and $\sigma_i^2$ as the $i$-th elements on the diagonal of $\mathbf{A}$ and $\Sigma$, respectively. As the $i$-th element of the noise vector $\mathbf{c}$ satisfies $\mathbb{E}[c_i^2] = (\theta_i^*)^2 + \sigma_i^2$, the expected loss of the iterates $\theta_t$ can be written as follows:

$$\mathcal{L}(\theta_t) = \mathbb{E}[\hat{\mathcal{L}}(\theta_t)] = \frac{1}{2} \sum_i a_i (\mathbb{E}[\theta_{t,i}]^2 + \mathbb{V}[\theta_{t,i}] + \sigma_i^2), \tag{3}$$

where $\theta_{t,i}$ represents the $i^{th}$ item of the parameter $\theta_t$. The limiting risk of SGD, Lookahead and Lookaround are compared in the following Proposition by unwrapping $\mathbb{E}[\theta_t]$ and $\mathbb{V}[\theta_t]$ in Equation 3.

**Proposition 1** (Steady-state risk). *Let $0 < \gamma < 1/L$ be the learning rate satisfying $L = \max_i a_i$. One can obtain that, in the noisy quadratic setup, the variance of the iterates obtained by SGD, Lookahead [49] and Lookaround converge to the following matrix:*

$$V_{SGD}^* = \frac{\gamma^2 \mathbf{A}^2 \Sigma^2}{\mathbf{I} - (\mathbf{I} - \gamma \mathbf{A})^2}, \tag{4}$$

$$V_{Lookahead}^* = \underbrace{\frac{\alpha^2 (\mathbf{I} - (\mathbf{I} - \gamma \mathbf{A})^{2k})}{\alpha^2 (\mathbf{I} - (\mathbf{I} - \gamma \mathbf{A})^{2k}) + 2\alpha(1 - \alpha)(\mathbf{I} - (\mathbf{I} - \gamma \mathbf{A})^k)}}_{\preccurlyeq \mathbf{I}, \, if \, \alpha \in (0,1)} V_{SGD}^*, \tag{5}$$

$$V_{Lookaround}^* = \underbrace{\frac{\alpha^2 (\mathbf{I} - (\mathbf{I} - \gamma \mathbf{A})^{2k}) + 2\alpha(1 - \alpha)(\mathbf{I} - (\mathbf{I} - \gamma \mathbf{A})^k)}{\alpha^2 (d\mathbf{I} - (d-1)(\mathbf{I} - \gamma \mathbf{A})^{2k})}}_{\preccurlyeq \mathbf{I}, \, if \, d \geqslant 3 \, and \, \alpha \in [1/2,1)} V_{Lookahead}^*, \tag{6}$$

*respectively, where $\alpha$ denotes the average weight factor of models with varying trajectory points, as described in [49].*

Proposition 1 implies that the steady-state variance matrix of Lookaround is element-wise smaller than those of SGD and Lookahead if $d \geqslant 3$ and $\alpha \in [1/2, 1)$. Moreover, Proposition 1 shows that increasing the number of data augmentation methods $d$ yields smaller limiting variance and thus lower expected loss (see equation 3), which guarantees good generalizability in real-world scenarios. The proof is deferred to Appendix B.

### 3.2.2 Convergence on Deterministic Quadratic function

In this section, we analyze the convergence of Lookaround and Lookahead in the noise-free quadratic setting, which is important for studying the convergence of optimization algorithms [13, 37, 44].

Convergence rate $\rho$ characterizes the speed in quadratic function at which the independent variable converges to the optimal solution, satisfying $||\theta_t - \theta*|| \leqslant \rho^t||\theta_0 - \theta*||$. In order to calculate the convergence rate, we model the optimization process, and treating this function as a linear dynamical system allows us to calculate the convergence rate, as in [30]. The value of the convergence rate will be determined by the eigenvalues of the dynamic transition equation, and we leave the calculation of this part to show in the Appendix B.2.

Given the emphasized significance of the condition number in paper [37], we conducted extensive experiments under a series of conditional numbers.

We evaluated the convergence rate of Lookaround, Lookahead, and Classical Momentum (CM) under different condition numbers from $10^1$ to $10^7$, which are shown in Figure 2. The blue dashed line represents the optimal convergence rate that an algorithm can achieve in the absence of any constraints. A value closer to the blue dashed line indicates a faster convergence rate. CM can achieve the optimal convergence rate on a critical condition number [30]. To the left of this critical point, there exist complex eigenvalues that correspond to oscillations. In this regime, CM exhibits a flat convergence rate, which is known as "under-damped" behavior [30]. We show that Lookaround and Lookahead are faster than CM in the under-damped regime. Lookaround converges faster and is more stable than Lookahead in the low condition num-

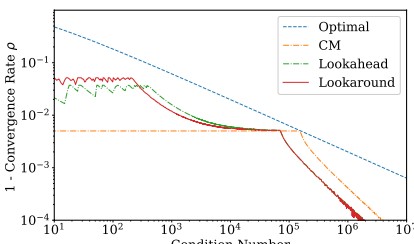

Figure 2: Convergence rate on quadratics of varying condition number. We fix the step $k = 20$ for Lookahead and Lookaround, and fix the CM factor $\beta = 0.99$.

ber case, and realizes comparable performance with Lookahead in the high condition number case. In subsequent experiments, we show that Lookaround achieves fast and stable convergence in training deep neural networks on popular benchmarks.

## 4 Experiments

In Section 4.1 and Section 4.2, we present experimental verification on different tasks, including both random initialization and finetuning tasks on both ViTs and CNNs architectures to validate the effectiveness of the Lookaround optimizer. In Section 4.4, we compare the proposed method with ensemble learning methods that require multiple models. In Section 4.5, we conduct ablation experiments on different components of the Lookaround method to explore their contributions. In Section 4.6, we analyze the parameter robustness of Lookaround and further investigate its performance in the micro-domain.

Within a single epoch, both the proposed Lookaround and the competitors undergo training on an identical times the data augmentations. With such a setup, we guarantee consistency in the data volume utilized by each method, thereby ensuring fair comparisons in terms of computation.

### 4.1 Random Initialization

Random initialization is a standard model initialization method in deep learning. In this subsection, we verify the performance of our algorithm on various networks with randomly initialized weights based on CIFAR and ImageNet datasets.

### 4.1.1 CIFAR 10 and CIFAR 100

We conduct our experiments on CIFAR10 [26] and CIFAR100 [26] datasets. Both CIFAR10 and CIFAR100 datasets have 60,000 images, 50,000 of which are used for training and 10,000 for validation. We use SGDM [36] as our baseline, and we compare our method with SWA [19], SWAD [3], Lookahead [49]. We validate different methods on multiple network architectures such as VGG19 [43], ResNet50 [17], ResNet101 [17], ResNet152 [17], ResNeXt50 [48] and all methods are

Table 1: Test set accuracy under training procedure with random initialized models. In this table, all models are trained for same amount of time at $32\times32$ resolution. (To provide a more comprehensive view of the data, we substitut the Top5 metric for CIFAR10 with the NLL loss.)

| | Method | VGG19 | | ResNet50 | | ResNet101 | | ResNet152 | | ResNext50 | |
|---|---|---|---|---|---|---|---|---|---|---|---|
| | | Top1 | NLL | Top1 | NLL | Top1 | NLL | Top1 | NLL | Top1 | NLL |
| CIFAR10 | SGDM | 93.92 | 0.26 | 95.96 | 0.16 | 96.16 | 0.16 | 96.28 | 0.16 | 95.72 | 0.17 |
| | SWA | **94.89** | **0.21** | 96.42 | 0.14 | 96.34 | 0.14 | 96.82 | **0.13** | 96.09 | 0.16 |
| | SWAD | 93.23 | 0.29 | 95.39 | 0.19 | 94.49 | 0.22 | 95.00 | 0.20 | 93.89 | 0.26 |
| | Lookahead | 94.72 | 0.23 | 96.38 | 0.16 | 96.61 | 0.15 | 96.46 | 0.15 | 96.47 | 0.15 |
| | Ours | 94.44 | 0.25 | **96.59** | **0.14** | **96.73** | **0.14** | **97.02** | 0.14 | **96.70** | **0.13** |
| | Method | VGG19 | | ResNet50 | | ResNet101 | | ResNet152 | | ResNext50 | |
| | | Top1 | Top5 | Top1 | Top5 | Top1 | Top5 | Top1 | Top5 | Top1 | Top5 |
| CIFAR100 | SGDM | 73.84 | 91.09 | 79.61 | 95.21 | 79.91 | 95.54 | 80.16 | 95.49 | 79.10 | 94.86 |
| | SWA | 73.62 | 90.89 | 80.17 | 95.56 | 80.53 | 95.39 | 80.86 | 95.43 | 79.14 | 94.96 |
| | SWAD | 73.37 | 89.93 | 80.19 | 95.46 | 79.92 | 95.09 | 80.00 | 95.36 | 79.27 | 95.17 |
| | Lookahead | 74.02 | 90.68 | 80.06 | 95.24 | 81.14 | 95.84 | 81.36 | 95.77 | 80.12 | 95.26 |
| | Ours | **74.29** | **91.15** | **81.60** | **95.99** | **81.97** | **96.15** | **82.22** | **96.40** | **81.14** | **96.12** |

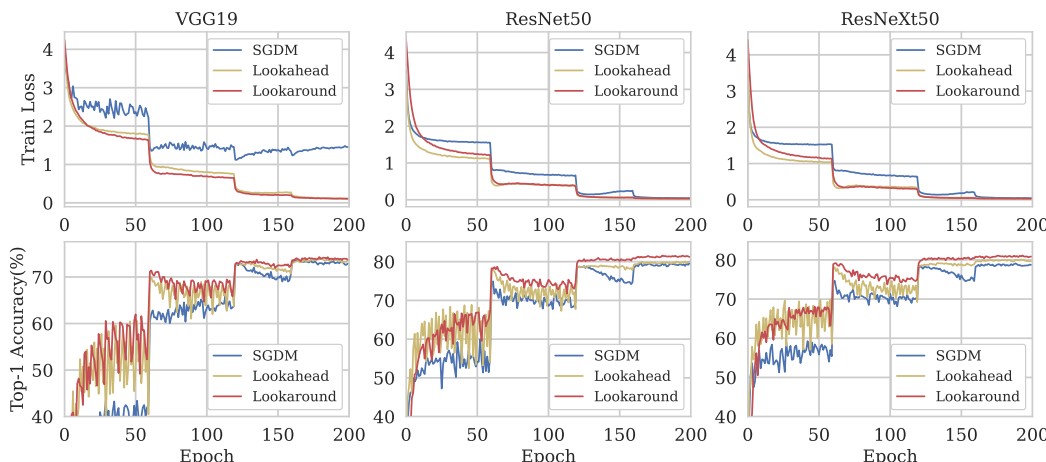

Figure 3: Training loss and Top-1 accuracy curves of various networks on CIFAR100 under different optimization methods.

performed on well-validated parameters. For each epoch in all experiments, we use the same mixed data for training using three kinds of data augmentation: random horizontal flip, random vertical flip, and RandAugment [5]. This ensures that the comparison between different methods is fair. For the training process, we use a discount factor of size 0.2 to decay the initial learning rate 0.1 at 60, 120, 160 epochs. Please refer to Appendix C.1.1 for more specific details.

We present our results in Figure 3 and Table 1, which indicates that our method achieves stable performance improvement on different networks. Compared to the SGDM optimizer, both Lookaround and Lookahead demonstrate faster convergence rates and higher accuracy. However, what's noteworthy is that, with training losses similar to Lookahead, Lookaround not only achieves a higher accuracy but also has better stability. This suggests that within the same training duration, we have found a more optimal optimization path. Meanwhile, although the SWA and SWAD methods adopt the same optimization process as the SGDM, they achieve higher performance via multiple samplings in the training trajectory combined with an average step. In contrast to SWA and SWAD, the multi-averaging Lookaround can achieve consistent and superior performance across different networks.

### 4.1.2 ImageNet

The ImageNet dataset is widely used to test the model performance. It has a training set of 1.28 million images and a validation set of 50,000 images with 1,000 classes. We train the model

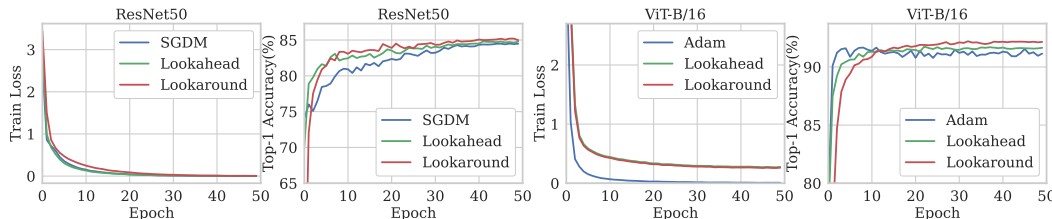

Figure 4: Training loss and Top-1 accuracy curves of ResNet50 (left) and ViT-B/16 (right) on CIFAR100 under different optimization methods.

on the training set and observe the model performance on the validation set. Based on standard hyperparameter settings, we train for 100 epochs and use a multi-step scheduler to adjust the learning rate on 30, 60 and 90 epochs with a discount factor of size 0.1. We present our results in Table 2. We get similar conclusions on the ImageNet as on CIFAR: higher test set accuracy and stronger generalization performance, which coincides with the theoretical results in the previous section.

Table 2: Top-1 accuracy of ResNet50 on ImageNet under different optimization methods.

| OPTIMIZER | TOP-1 | TOP-5 |
|---|---|---|
| SGDM | 75.97 | 92.89 |
| SWA | 76.78 | 93.18 |
| LOOKAHEAD | 76.52 | 93.11 |
| LOOKAROUND | **77.32** | **93.29** |

## 4.2 Finetuning

The transfer learning technique, which involves pretraining a model on a large related dataset and then finetuning on a smaller target dataset of interest, has become a widely used and powerful approach for creating high-quality models across various tasks.

To explore the scalability of our Lookaround method, we apply our method to scenarios that require pre-training, including CNN or ViT [7] architecture. Unlike the previous experiments, we adopt more appropriate hyperparameters for its training in the ViT experiments, including the Adam optimizer and cosine annealing scheduler structure. Our results are shown in Table 3 and Figure 4. Under the ResNet architecture, our proposed Lookaround method remains stable and performs better, while the SGDM method shows greater fluctuations.

In contrast, the Lookaround methods based on weight averaging do not produce such a phenomenon, which partly demonstrates the hypothesis that average weight can reduce the convergence variance. In our comparison experiments with the other optimizations, we found that the results of ViT-B/16 under different optimizers were better than those reported in the original paper [7]. This is mainly due to our use of more data augmentation during the training process, aiming for a fair comparison with Lookaround. This result also indicates that Lookaround can still achieve better performance under improved baseline conditions.

Table 3: The test set accuracy and the NLL loss under training procedure with pretrained models. In this table, all models are fine-tuned for the same amount of time at 224×224 resolution. The results of Adam[†] using ViT-B/16 are taken from reference [7].

| Backbone | Method | CIFAR10 | | | CIFAR100 | | |
|---|---|---|---|---|---|---|---|
| | | **Top-1** | **NLL** | **#param**. | **Top-1** | **NLL** | **#param**. |
| ResNet50 | SGDM | 97.55 | 0.109 | 23.52M | 84.50 | 0.692 | 23.71M |
| | SWA | 97.48 | 0.105 | 23.52M | 84.81 | 0.668 | 23.71M |
| | Lookahead | 97.65 | 0.103 | 23.52M | 84.78 | 0.676 | 23.71M |
| | Ours | **97.82** | **0.099** | 23.52M | **85.20** | **0.658** | 23.71M |
| ViT-B/16 | Adam[†] | 98.13 | - | 85.65M | 87.13 | - | 85.72M |
| | Adam | 98.34 | 0.060 | 85.65M | 91.55 | 0.298 | 85.72M |
| | SWA | 98.47 | 0.049 | 85.65M | 91.32 | 0.304 | 85.72M |
| | Lookahead | 98.51 | 0.047 | 85.65M | 91.76 | 0.280 | 85.72M |
| | Ours | **98.71** | **0.041** | 85.65M | **92.21** | **0.267** | 85.72M |

Table 4: The test set accuracy under SAM and Lookaround optimization using ResNet50 or ViT-B/16. The results of SAM[†] using ResNet50 are taken from [7].

| Backbone | Method | resolution | pretrain | CIFAR10 | CIFAR100 |
|---|---|---|---|---|---|
| ResNet50 | SAM[†] | 32 | - | - | 79.10 |
| | Lookaround | 32 | - | **96.59** | **81.60** |
| | Lookaround + SAM | 32 | - | 96.38 | 79.79 |
| ResNet50 | SAM | 224 | ✓ | 97.65 | 85.97 |
| | Lookaround | 224 | ✓ | 97.82 | 85.20 |
| | Lookaround + SAM | 224 | ✓ | **97.88** | **86.21** |
| ViT-B/16 | SAM | 224 | ✓ | 98.51 | 92.39 |
| | Lookaround | 224 | ✓ | **98.71** | 92.21 |
| | Lookaround + SAM | 224 | ✓ | 98.67 | **92.54** |

## 4.3 Compared with Sharpness-Aware Minimization

We draw a comparison between Lookaround and Sharpness-Aware Minimization (SAM) [10], an algorithm designed to improve generalization by steering model parameters towards flatter loss regions.[1] The comparative results are presented in Table 4.

The results seem to indicate that, under the default parameters of SAM, this method is more suitable for higher resolutions, while Lookaround achieves the best performance at lower resolutions. For the finetuning, neither SAM nor Lookaround consistently emerges as the top choice when used individually. However, when combined, they often enhance model performance significantly. Thus, for higher accuracy, we recommend using both the SAM and Lookaround optimizers concurrently. Both are orthogonal to each other. By combining the two methods, we can achieve superior performance.

## 4.4 Compared with Ensemble Method

Ensemble methods are traditionally used to further improve inference performance by combining the outputs of multiple different models. We compare our Lookaround method with classical Ensemble methods (Logit Ensemble and Snapshot Ensemble [18]), and the results are shown in Table 5.

The performance of the single model obtained by our training exceeds the performance of the ensemble under multiple models. Moreover, the model obtained by our method is not a simple superposition of Ensemble models. We take the single model obtained by us with the three models obtained by Logit Ensemble to Logit Ensemble, and get a more significant performance improvement. This indicates that the model we obtained is orthogonal to the three models obtained by Logit Ensemble and can be combined to produce better performance.

Table 5: Performance compared with the ensemble methods (without pretrained weights) on CIFAR100 dataset. "Ours + Logit Ensemble" means that we take the model obtained by our method and the three models in Logit Ensemble to Logit Ensemble.

| Method | ResNet50 | ResNet101 | ResNet152 |
|---|---|---|---|
| Logit Ensemble | 81.16 | 81.92 | 81.89 |
| SnapShot Ensemble | 79.88 | 80.63 | 80.23 |
| Ours (Lookaround) | 81.60 | 81.97 | 82.22 |
| Ours + Logit Ensemble | **83.43** | **83.35** | **83.53** |

## 4.5 Ablation Study

In this subsection, we investigate the impact of different components of Lookaround on its performance, including data Augmentation methods and weight averaging. As shown in Table 6,

Table 6: Ablation Study of Data Augmentation (DA) and Weight Averaging (WA) in the proposed Lookaround using ResNet50.

| Dataset | DA | WA | Acc (%) | Dataset | DA | WA | Acc (%) |
|---|---|---|---|---|---|---|---|
| CIFAR10 | - | - | 95.84 | CIFAR100 | - | - | 78.81 |
| | ✓ | - | 95.96 | | ✓ | - | 79.61 |
| | - | ✓ | 95.79 | | - | ✓ | 79.49 |
| | ✓ | ✓ | **96.59** | | ✓ | ✓ | **81.60** |

---

[1]Intriguingly, SAM is proved to exhibit dynamics akin to a decentralized weight averaging algorithm [51].

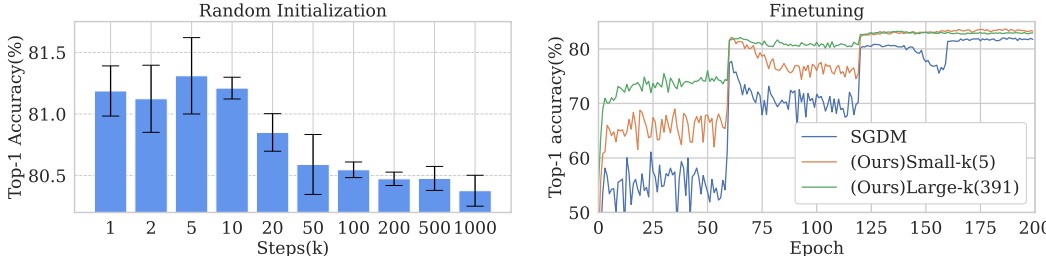

Figure 5: Top-1 accuracy of ResNet50 on CIFAR100 dataset with different steps $k$. **(Left)** Error bar of three different random seeds with random initialization weight. **(Right)** Top-1 accuracy of two different $k$ values with pretrained weight.

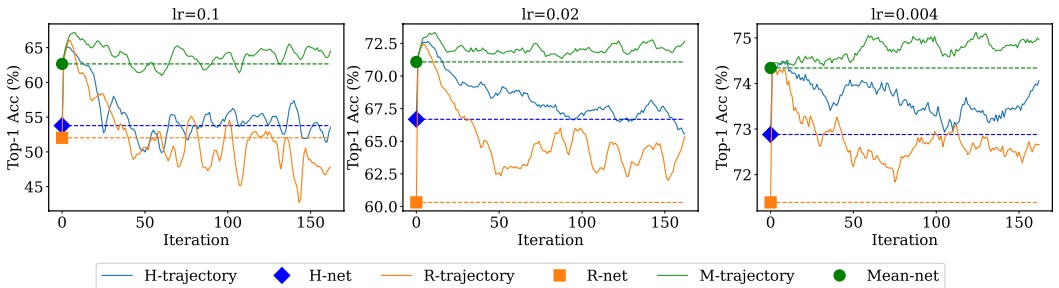

Figure 6: Accuracy curve of sub-model state update on the test set with different learning rates using ResNet50 on CIFAR100. At position 0 on the horizontal axis, H-net (trained by randomly horizontally flipped data) and R-net (trained by RandAugment data) are averaged to Mean-net. Subsequently, during 160 weight iterations, H-net and R-net continue to be trained with their respective data, while Mean-net represents the model obtained by averaging the weights at each itreation.

when we consider Data Augmentation (DA) or Weight Averaging (WA) independently, their improvement effects may be relatively small. However, when we combine them, the synergistic effect between them can lead to more significant overall improvement. Specifically, we found that the improvement effects of DA and WA show a nonlinear trend in the interaction case, indicating a certain synergistic effect between them. Under the interaction, Data augmentation guides the model into different loss basins, and Weight Averaging guides different models into lower loss basins, resulting in higher performance improvement and a qualitative leap.

### 4.6 Additional Analysis

**Robustness of the Parameters.** For the number of data augmentations, we train the model with different numbers of data augmentations, and the results are shown in Table 7. The results indicate that an increase in the number of data augmentations results in improved network performance. In

Table 7: Top-1 accuracy (%) of different data augmentation (DA) number by using ResNet50 on CIFAR100 dataset.

| # of DA | 1 | 2 | 3 | 4 | 5 | 6 |
|---|---|---|---|---|---|---|
| Top-1 (%) | 78.20 | 80.82 | 81.60 | 81.19 | 81.74 | **82.02** |
| Top-5 (%) | 94.50 | 95.19 | 95.99 | 95.65 | 95.85 | **96.02** |

order to reduce the training time, only three data augmentation methods are selected in this paper for experimental verification at different levels. Note that the methods compared with ours use the same incremental data set, so the comparison is fair.

For different choices of $k$, the number of around steps, we select different intervals for full verification, and the results are shown in Figure 5 (Left). Let us look at Large-k (391) versus Small-k(5) in Figure 5 (Right) (391 is determined by the number of batches in a single epoch). We find that large steps (green line) can achieve higher accuracy and stability in the early stage of training, while small steps can achieve higher accuracy in the later stage. In future research, we may explore a learning rate scheduler suitable for Lookaround to help it achieve the best performance.

We visualize the training trajectory accuracy of the sub-models in Lookaround in Figure 6. Under different learning rates, the model, after averaging the weights, can maintain a good baseline and continuously improve the performance of the model under the low variance, and the sub-models also obtain more beneficial training effects due to the weight averaging.

## 5    Conclusion

In this paper, we present the Lookaround optimizer, an SGD-based optimizer that employs a diversity-in-locality exploration strategy to find solutions with high generalization ability in the loss basin. We theoretically prove that Lookaround can obtain lower variance and faster convergence speed. Experimental results on CIFAR and ImageNet and across various network backbones show that, our proposed Lookaround significantly improves the training stability and achieves better generalization performance comparable to the single-trajectory WA optimizers and ensemble methods.

**Limitation.**   Lookaround is limited by the additional cost of network searching using different data augmentation, resulting in a longer training time proportional to the the number of trained networks. Notably, Lookaround incurs the same inference time as other optimization methods. Considering the significantly superior performance, we believe the extra time consumption is worthwhile in many real-world scenarios where the training time is not so concerned as the inference time. Moreover, training time can be reduced due to the parallelizability of the "around step" mechanism in Lookaround.

## Acknowledgements

This work is supported by the National Key Research and Development Project (Grant No: 2022YFB2703100), National Natural Science Foundation of China (62106220, 61976186, U20B2066), and Ningbo Natural Science Foundation (2021J189) and the advanced computing resources provided by the Supercomputing Center of Hangzhou City University. We sincerely thank the anonymous reviewers for their valuable comments on the work.

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

# A    Exploration in the Loss Landscape

To demonstrate the effectiveness of Lookaround during training, we set up an experiment and visualize the loss landscape of the models under different data augmentations in Figure 7. By observing the loss landscape, we can gain a clearer understanding of the role played by the weight averaging at different stages during the training process.

**Training Process and Parameter Settings.**    We first train a ResNet50 on the CIFAR100 dataset. The learning rate is initialized to 0.1 and decay at 60, 120, and 160 epochs using a MultiStepLR scheduler with a decay factor 0.2. The batch size is set to 128, we use stochastic gradient descent with momentum to optimize the model and use random crops and random vertical flips augmentation to enhance the training datasets. We use model checkpoints at epochs 50, 110, and 150 as our three pretrained models(V-network). The three pretrained models correspond to learning rates of 0.1, 0.02 and 0.004, respectively. Using these pretrained models as the starting point, we finetune the model with 1, 10, 100, 1000, 10000 iterations under the corresponding learning rate and the setting of random horizontal flipping (H-network) or RandAugment (R-network).

**Visualization Method.**    We use the visualization method in Garipov et al. [12] to plot the loss landscape. In this method, the weights of the three models are flattened respectively as one-dimensional vectors $w_v, w_h, w_r$, and then two orthogonal vectors are calculated between the three vectors as the X-axis direction and the Y-axis direction: $u = (w_h - w_v), v = (w_r - w_v, w_h - w_v)/\|w_h - w_v\|^2 \cdot (w_h - w_v)$. Then the normalized vectors $\hat{u} = u/\|u\|, \hat{v} = v/\|v\|$ foam an orthonormal basis in the plane contain $w_v, w_h, w_r$. Then a point P with coordinates $(x, y)$ in the plane would be given by $P = w_v + x \cdot \hat{u} + y \cdot \hat{v}$.

**Discussion and Inspiring.**    Under different learning rates and different around steps $k$, Lookaround has the tendency to lead the model trained on different data augmentation to the near-constant loss manifold. In such circumstances, the "average step" can lead the model into the center of the loss basin to get a lower test loss. However, weight averaging does not necessarily work in all cases. For example, the network obtained after weight averaging gets a larger loss under a large learning rate with a large around step (e.g., $lr = 0.1, k = 10000$). In this case, the model is located on a peak between different basins rather than in the center of a basin. Moreover, in the case of around step $k = 1$, the weight averaging also does not achieve better performance. Nevertheless, such extreme cases do not prevent weight averaging from being a useful tool to speed up the training process. The center of the basin in loss landscape, which requires multi-step gradient descent to reach, can be reached by only one weight averaging step. At other learning rates and around steps $k$, the models after weight averaging all result in a lower loss than the individual model. Such phenomena encourage us to explore more methods to find more optimal solutions in the loss landscape in the future.

# B    Steady-state and Convergence Analysis of Lookaround

We use quadratic functions to analyze the steady-state and convergence analysis of Lookaround. First, we present the proof of Proposition 2.

**Proposition 2** (Steady-state risk). *Let $0 < \gamma < 1/L$ be the learning rate satisfying $L = \max_i a_i$. One can obtain that in the noisy quadratic setup, the variances of SGD, Lookahead [49] and Lookaround converge to the following fixed matrix:*

$$V_{SGD}^* = \frac{\gamma^2 \mathbf{A}^2 \Sigma^2}{\mathbf{I} - (\mathbf{I} - \gamma \mathbf{A})^2}, \tag{7}$$

$$V_{Lookahead}^* = \underbrace{\frac{\alpha^2(\mathbf{I} - (\mathbf{I} - \gamma \mathbf{A})^{2k})}{\alpha^2(\mathbf{I} - (\mathbf{I} - \gamma \mathbf{A})^{2k}) + 2\alpha(1 - \alpha)(\mathbf{I} - (\mathbf{I} - \gamma \mathbf{A})^k)}}_{\preccurlyeq \mathbf{I}, \, if \, \alpha \in (0,1)} V_{SGD}^*, \tag{8}$$

$$V_{Lookaround}^* = \underbrace{\frac{\alpha^2(\mathbf{I} - (\mathbf{I} - \gamma \mathbf{A})^{2k}) + 2\alpha(1 - \alpha)(\mathbf{I} - (\mathbf{I} - \gamma \mathbf{A})^k)}{\alpha^2(d\mathbf{I} - (d - 1)(\mathbf{I} - \gamma \mathbf{A})^{2k})}}_{\preccurlyeq \mathbf{I}, \, if \, d \geqslant 3 \, and \, \alpha \in [1/2,1)} V_{Lookahead}^*. \tag{9}$$

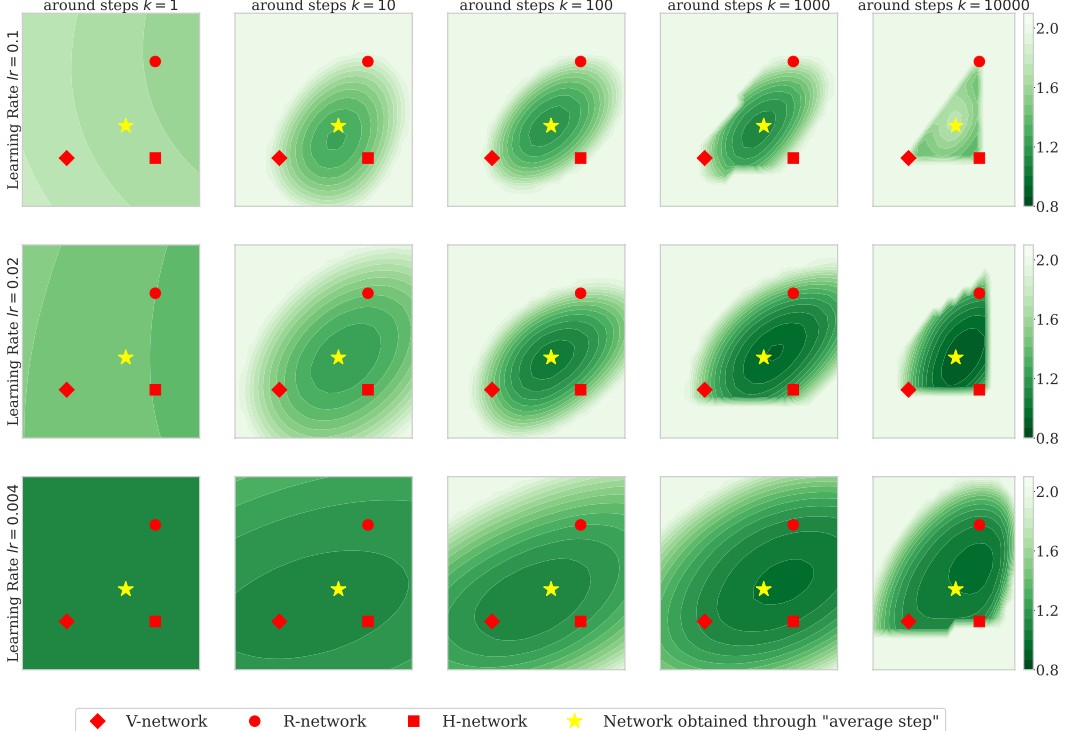

Figure 7: The test loss landscape of ResNet50 on CIFAR100. The diamond block (V-network) represents the pretrained model trained with random vertical flipping. Then we can use random horizontal flipping and RandAugment to finetune V-network to get H-network and R-network.

*respectively, where $\alpha$ denotes the average weight factor of models with varying trajectory points, as described in [49].*

From Wu et al. [47], we can obtain the following dynamics of SGD with learning rate $\gamma$:

$$\mathbb{E}[\theta^{(t+1)}] = (\mathbf{I} - \gamma\mathbf{A})\,\mathbb{E}[\theta^{(t)}],$$
$$\mathbb{V}[\theta^{(t+1)}] = (\mathbf{I} - \gamma\mathbf{A})^2\,\mathbb{V}[\theta^{(t)}] + \gamma^2\mathbf{A}^2\Sigma.$$

**Lemma 1.** *The expectation and variance of lookaround have the following iterates:*

$$\mathbb{E}[\phi^{(t+1)}] = (\mathbf{I} - \gamma\mathbf{A})^k\,\mathbb{E}[\phi^{(t)}], \tag{10}$$

$$\mathbb{V}[\phi^{(t+1)}] = \frac{d-1}{d}(\mathbf{I} - \gamma\mathbf{A})^{2k}\,\mathbb{V}[\phi^{(t)}] + \frac{\gamma^2\mathbf{A}^2\Sigma(\mathbf{I} - (\mathbf{I} - \gamma\mathbf{A})^{2k})}{d(\mathbf{I} - (\mathbf{I} - \gamma\mathbf{A})^2)}. \tag{11}$$

$\phi_t$ yields $\phi_{t+1}$ by performing an around step and an average step.

*Proof.* The expected iterate follows from SGD:

$$\mathbb{E}[\phi^{t+1}] = \mathbb{E}[\frac{1}{d}\sum_i \theta_{t,i,k}] = \sum_i \frac{1}{d}\mathbb{E}[\theta_{t,i,k}]$$
$$= \sum_i \frac{1}{d}(\mathbf{I} - \gamma\mathbf{A})^k\,\mathbb{E}[\theta_{t,i,0}] = (\mathbf{I} - \gamma\mathbf{A})^k\,\mathbb{E}[\phi^t].$$

For the variance of $\phi^{t+1}$, we can break it down into two parts as follows:

$$\mathbb{V}[\phi^{t+1}] = \mathbb{V}[\frac{1}{d}\sum_i \theta_{t,i,k}] = \sum_i^d \frac{1}{d^2}\mathbb{V}[\theta_{t,i,k}] + \sum_{i\neq j,1\leqslant i,j\leqslant d}\frac{1}{d^2}\mathrm{cov}(\theta_{t,i,k},\theta_{t,j,k}).$$

The covariance of the different models can be calculated in the following way:

$$
\begin{aligned}
\mathrm{cov}(\theta_{t,i,k}, \theta_{t,j,k}) &= \mathbb{E}[\theta_{t,i,k}\theta_{t,j,k}] - \mathbb{E}[\theta_{t,i,k}]\,\mathbb{E}[\theta_{t,j,k}] \\
&= \mathbb{E}[(\mathbf{I} - \gamma\mathbf{A})^{2k}(\phi^t)^2] - (\mathbf{I} - \gamma\mathbf{A})^{2k}\,\mathbb{E}[\phi^t]^2 \\
&= (\mathbf{I} - \gamma\mathbf{A})^{2k}\,\mathbb{V}[\phi^t].
\end{aligned}
$$

After permuting and regrouping again, we can obtain the iterate with respect to the variance.

$$
\begin{aligned}
\mathbb{V}[\phi^{t+1}] &= \sum_{i \neq j, 1 \leqslant i,j \leqslant d} \frac{1}{d^2}\mathrm{cov}(\theta_{t,i,k}, \theta_{t,j,k}) + \sum_i^d \frac{1}{d^2}\,\mathbb{V}[\theta_{t,i,k}] \\
&= \frac{1}{d^2}(d^2 - d)(\mathbf{I} - \gamma\mathbf{A})^{2k}\,\mathbb{V}[\phi^t] + \frac{1}{d}\Big[\sum_{i=0}^{k-1}(\mathbf{I} - \gamma\mathbf{A})^{2i}\gamma^2\mathbf{A}^2\Sigma\Big] \\
&= \frac{d-1}{d}(\mathbf{I} - \gamma\mathbf{A})^{2k}\,\mathbb{V}[\phi^t] + \frac{\gamma^2\mathbf{A}^2\Sigma(\mathbf{I} - (\mathbf{I} - \gamma\mathbf{A})^{2k})}{d(\mathbf{I} - (\mathbf{I} - \gamma\mathbf{A})^2)}.
\end{aligned}
$$

The proof is now complete. $\qquad\square$

**Remark.** From Equation 10, the expectation term for $\phi$ in Lookaround eventually converges to 0, as does Lookahead and SGD.

From Zhang et al. [49], we have the following analysis about the variance fixed point of lookahead with learning rate $\gamma$ and weight factor $\alpha$, which represents the average weight factor of models with different trajectory points in the Lookahead optimizer, which is generally $(0, 1)$:

$$
V^*_{Lookahead} = \frac{\alpha^2(\mathbf{I} - (\mathbf{I} - \gamma\mathbf{A})^{2k})}{\alpha^2(\mathbf{I} - (\mathbf{I} - \gamma\mathbf{A})^{2k}) + 2\alpha(1 - \alpha)(\mathbf{I} - (\mathbf{I} - \gamma\mathbf{A})^k)}V^*_{SGD}. \tag{12}
$$

We now derive the fixed point of the variance, proceed with the proof of Proposition 1:

$$
V^*_{Lookaround} = \frac{\alpha^2(\mathbf{I} - (\mathbf{I} - \gamma\mathbf{A})^{2k}) + 2\alpha(1 - \alpha)(\mathbf{I} - (\mathbf{I} - \gamma\mathbf{A})^k)}{\alpha^2(d\mathbf{I} - (d - 1)(\mathbf{I} - \gamma\mathbf{A})^{2k})}V^*_{Lookahead}.
$$

*Proof.*

$$
\begin{aligned}
V^*_{Lookaround} &= \frac{d-1}{d}(\mathbf{I} - \gamma\mathbf{A})^{2k}V^*_{Lookaround} + \frac{\gamma^2\mathbf{A}^2\Sigma(\mathbf{I} - (\mathbf{I} - \gamma\mathbf{A})^{2k})}{d(\mathbf{I} - (\mathbf{I} - \gamma\mathbf{A})^2)}. \\
\Rightarrow V^*_{Lookaround} &= \frac{1}{\mathbf{I} - \frac{d-1}{d}(\mathbf{I} - \gamma\mathbf{A})^{2k}}\frac{\gamma^2\mathbf{A}^2\Sigma(\mathbf{I} - (\mathbf{I} - \gamma\mathbf{A})^{2k})}{d(\mathbf{I} - (\mathbf{I} - \gamma\mathbf{A})^2)} \\
&= \frac{\gamma^2\mathbf{A}^2\Sigma[\mathbf{I} - (\mathbf{I} - \gamma\mathbf{I})^{2k}]}{[d\mathbf{I} - (d - 1)(\mathbf{I} - \gamma\mathbf{A})^{2k}][\mathbf{I} - (\mathbf{I} - \gamma\mathbf{A})^2]} \\
&= \frac{\mathbf{I} - (\mathbf{I} - \gamma\mathbf{A})^{2k}}{d\mathbf{I} - (d - 1)(\mathbf{I} - \gamma\mathbf{A})^{2k}}V^*_{SGD}.
\end{aligned}
$$

According to Equation 12, we can deduce that

$$
\begin{aligned}
V^*_{Lookaround} &= \frac{\mathbf{I} - (\mathbf{I} - \gamma\mathbf{A})^{2k}}{d\mathbf{I} - (d - 1)(\mathbf{I} - \gamma\mathbf{A})^{2k}}V^*_{SGD} \\
&= \frac{\alpha^2(\mathbf{I} - (\mathbf{I} - \gamma\mathbf{A})^{2k}) + 2\alpha(1 - \alpha)(\mathbf{I} - (\mathbf{I} - \gamma\mathbf{A})^k)}{\alpha^2(d\mathbf{I} - (d - 1)(\mathbf{I} - \gamma\mathbf{A})^{2k})}V^*_{Lookahead}.
\end{aligned}
$$

The proof is now complete. $\qquad\square$

## B.1 Comparing the dynamics of Lookahead

We now proceed with the proof for the range of constraints variable $\alpha$ in Equation 9. When $d \geqslant 3$, and $\alpha \in [0.5, 1)$, the Lookaround method can obtain a smaller variance than the Lookahead method:

*Proof.* Let $B = (\mathbf{I} - \gamma\mathbf{A})^k$, due to $0 < \gamma < 1/L, L = \max_i a_i$, so we can have $B \preccurlyeq \mathbf{I}$. Substituting the matrix B into the expressions for the variance fixed point relation of Lookaround and Lookahead:

$$V^*_{Lookaround} = \frac{\alpha^2(\mathbf{I} - (\mathbf{I} - \gamma\mathbf{A})^{2k}) + 2\alpha(1-\alpha)(\mathbf{I} - (\mathbf{I} - \gamma\mathbf{A})^k)}{\alpha^2(d\mathbf{I} - (d-1)(\mathbf{I} - \gamma\mathbf{A})^{2k})} V^*_{Lookahead}$$

$$= \frac{\mathbf{I} - B^2 + 2\frac{1-\alpha}{\alpha}(\mathbf{I} - B)}{d\mathbf{I} - (d-1)B^2} V^*_{Lookahead}$$

$$= \frac{\frac{2-\alpha}{\alpha}\mathbf{I} - B^2 - \frac{2-2\alpha}{\alpha}B}{d\mathbf{I} - (d-1)B^2} V^*_{Lookahead}.$$

Let the coefficient matrix be denoted as C, when $d \geqslant 3$, for each diagonal element $C_{ii}$ of C, we can scale the denominator of this expression as follows:

$$C_{ii} \leqslant \frac{\frac{2-\alpha}{\alpha} - B_{ii}^2 - \frac{2-2\alpha}{\alpha}B_{ii}}{3 - 2B_{ii}^2},$$

Then, we can derive the range of $\alpha$ by restricting the right-hand side expression as follows:

$$\frac{\frac{2-\alpha}{\alpha} - B_{ii}^2 - \frac{2-2\alpha}{\alpha}B_{ii}}{3 - 2B_{ii}^2} \leqslant 1,$$

As $0 \leqslant B_{ii} \leqslant 1$, we can multiply both sides of the inequality by the denominator, then rearrange and combine like terms to obtain the following form:

$$B_{ii}^2 - \frac{2 - 2\alpha}{\alpha}B_{ii} + \frac{2 - 4\alpha}{\alpha} \leqslant 0.$$

Skipping the detailed steps, we can obtain $\alpha \geqslant 0.5$ by solving the quadratic equation. Therefore, in the case where $\alpha \in [0.5, 1)$ and $d \geqslant 3$ ($\alpha < 1$ is subject to Lookahead's settings), the coefficient matrix $C \preccurlyeq \mathbf{I}$, so the convergence speed of Lookaround is slower than Lookahead. $\square$

## B.2 Deterministic quadratic convergence

Our method samples data from multiple data augmentation strategies, which can be analogously viewed as averaging the sampling of historical trajectories during the convergence analysis of quadratic functions. Thus, for weight averaging, we select model points that align with each point in the $k$-step trajectory. From this perspective, we examine and compare the convergence rates of Lookaround and Lookahead.

We first show the state transition equation for the classical momentum method in quadratic functions:

$$\mathbf{v}_{t+1} = \beta\mathbf{v}_t - \nabla_\theta f(\theta_t) = \beta\mathbf{v}_t - \mathbf{A}\theta_t, \tag{13}$$

$$\theta_{t+1} = \theta_t + \gamma\mathbf{v}_{t+1} = \gamma\beta\mathbf{v}_t + (\mathbf{I} - \gamma\mathbf{A})\theta_t. \tag{14}$$

Here, $\mathbf{v}$ stands for the momentum term. We can generalize this to matrix form:

$$\begin{bmatrix} \mathbf{v}_{t+1} \\ \theta_{t+1} \end{bmatrix} = \begin{bmatrix} \beta & -\mathbf{A} \\ \gamma\beta & \mathbf{I} - \gamma\mathbf{A} \end{bmatrix} \begin{bmatrix} \mathbf{v}_t \\ \theta_t \end{bmatrix}.$$

Thus, given the initial $\theta_0$, we can obtain the convergence rate with respect to $\theta$ by the maximum eigenvalue of the matrix. Referring to Zhang et al. [49] and Lucas et al. [30], we obtain the state transition matrix regarding our algorithm as follows:

$$\begin{bmatrix} \boldsymbol{\theta}_{t,0} \\ \boldsymbol{\theta}_{t-1,k} \\ \vdots \\ \boldsymbol{\theta}_{t-1,1} \end{bmatrix} = LB^{(k-1)}T \begin{bmatrix} \boldsymbol{\theta}_{t-1,0} \\ \boldsymbol{\theta}_{t-2,k} \\ \vdots \\ \boldsymbol{\theta}_{t-2,1} \end{bmatrix},$$

where L, B and T denote the average weight matrix, the single-step transfer matrix and the position transformation matrix respectively:

$$L = \begin{bmatrix} \frac{1}{k+1}I & \frac{1}{k+1}I & \cdots & \frac{1}{k+1}I & \frac{1}{k+1}I \\ I & 0 & \cdots & \cdots & 0 \\ 0 & I & \ddots & \ddots & \vdots \\ \vdots & \ddots & \ddots & 0 & \vdots \\ 0 & \cdots & 0 & I & 0 \end{bmatrix},$$

$$B = \begin{bmatrix} (1+\beta)I - \eta A & -\beta I & 0 & \cdots & 0 \\ I & 0 & \cdots & \cdots & 0 \\ 0 & I & \ddots & \ddots & \vdots \\ \vdots & & \ddots & \ddots & 0 & \vdots \\ 0 & & \cdots & 0 & I & 0 \end{bmatrix},$$

$$T = \begin{bmatrix} I - \eta A & \beta I & -\beta I & 0 & \cdots & 0 \\ I & 0 & \cdots & \cdots & 0 & \vdots \\ 0 & I & \ddots & \cdots & \vdots & \vdots \\ \vdots & & \ddots & \ddots & 0 & \vdots & \vdots \\ \vdots & & \cdots & 0 & I & 0 & 0 \\ 0 & & \cdots & 0 & 0 & I & 0 \end{bmatrix}.$$

After specifying the appropriate parameters and performing matrix multiplication to obtain the state transition matrix, the convergence rate $\rho$ can be obtained by calculating the largest eigenvalue of the matrix. Note that since this linear dynamical system corresponds to $k$ updates, we finally have to compute the $k^{th}$ root of the eigenvalues to recover the correct convergence bounds.

# C Experimental Detail

## C.1 Random Initialization

### C.1.1 CIFAR10 and CIFAR100

**Data augmentation details.** For the CIFAR10 dataset, we use [RandomCrop + *] for data augmentation, and for the CIFAR100 dataset, we use [RandomCrop + * + RandomRotation] for data augmentation. * can be replaced by three different data augmentation methods of random horizontal flip, random vertical flip, or RandAugment [4].

**Training details.** For the CIFAR10 and CIFAR100 datasets, we have applied some common settings. The initial learning rate is set to 0.1, and the batch size is set to 128. Additionally, a warm-up phase of 1 epoch is implemented. Subsequently, different learning rate schedulers are used based on the specific dataset. For the CIFAR100 dataset, we utilize the MultiStepLR scheduler. The learning rate is decayed at the 60, 120, and 160 epochs, with a decay factor of 0.2. For the CIFAR10 dataset, we employ the CosineAnnealingLR learning rate scheduler. In comparison with other optimizers or optimization methods, we use the default settings of the other methods in the corresponding papers.

### C.1.2 ImageNet.

For the ImageNet dataset, our data augmentation strategy involves [RandomResizedCrop + * + RandomRotation]. Here, * can be substituted by one of the three data augmentation methods: random horizontal flip, random vertical flip, or RandAugment [4]. We adopt the following training settings: an initial learning rate of 0.1 and a batch size of 256. The model undergoes training for a total of 100 epochs. Additionally, we make use of the MultiStepLR scheduler, which decays the learning rate by a factor of 0.1 at the 30th, 60th, and 90th epochs.

## C.2 Finetuning

In this stage, all images are resized to 224 x 224 pixels to match the input size of the pretrained model. All models use the ImageNet-1k pretrained weights sourced from the PyTorch library. Throughout the finetuning process, we employed three data augmentation methods: random horizontal flip, RandAugment, and AutoAugment.

For the training of ResNet50, we performed a grid search as shown in Table 8 to determine the optimal learning rate and weight decay. Every training process lasted for 50 epochs with a batch size

Table 8: Hyperparameter search for finetuning on ResNet50.

| Dataset | Epochs | Base LR | Weight Decay |
|---|---|---|---|
| CIFAR10 | 50 | {0.001, 0.003, 0.01, 0.03} | {0.0001, 0.00005, 0.00001} |
| CIFAR100 | 50 | {0.001, 0.003, 0.01, 0.03} | {0.0001, 0.00005, 0.00001} |

Table 9: Comparison between different training strategies with ResNet18 on Stanford Cars dataset. Here "H" denotes random horizontal flip data augmentation. "R" denotes RandAugment data augmentation. "A" denotes AutoAugment data augmentation. "Logit Ensemble + SWA" denotes a logit ensemble approach that employs SWA models, which have been acquired through various data augmentation techniques. "T-Budget" and "I-Budget" respectively represent Training budget and Inference budget.

| Method | Data Augmentation Strategy | Top-1 Acc | Top-5 Acc | T-Budget | I-Budget |
|---|---|---|---|---|---|
| SGDM | "H" | 86.57 | 97.52 | 1x | 1x |
| SWA | "H" | 87.60 | 98.21 | 1x | 1x |
| SGDM | "R" | 87.30 | 98.01 | 1x | 1x |
| SWA | "R" | 88.00 | 98.21 | 1x | 1x |
| SGDM | "A" | 87.63 | 98.05 | 1x | 1x |
| SWA | "A" | 88.11 | 98.15 | 1x | 1x |
| Logit Ensemble | "H" + "R" | 89.24 | 98.44 | 2x | 2x |
| Lookaround | "H" + "R" | 89.35 | 98.47 | 2x | 1x |
| Logit Ensemble | "H" + "R" + "A" | 90.19 | 98.67 | 3x | 3x |
| Lookaround | "H" + "R" + "A" | 90.76 | 98.78 | 3x | 1x |
| Logit Ensemble+SWA | "H" + "R" + "A" | 90.39 | 98.80 | 3x | 3x |
| **Lookaround+SWA** | **"H" + "R" + "A"** | **91.04** | **98.86** | **3x** | **1x** |

set to 128. For the training of ViT-B/16, we used the Adam optimizer, starting with a learning rate of 0.00001. We set $\beta_1$ to 0.9, $\beta_2$ to 0.999, and applied a weight decay of 0.01. To optimize memory usage, the batch size was set to 64.

### C.3 Compared with Sharpness-Aware Minimization

We adopt the default hyperparameters from the origin paper of SAM [10] and combine them with the optimal hyperparameters found during random initialization or finetuning to conduct comparative experiments between SAM and Lookaround.

### C.4 Compared with Ensemble Method

We compare Lookaround with Logit Ensemble and Snapshot Ensemble [18]. In the setting of Logit Ensemble, we train multiple models separately using different data augmentation methods, and then average the outputs of these models for prediction. However, this approach requires more inference time. In the setting of Snapshot Ensemble, we use the CosineAnnealingWarmRestarts learning rate scheduler to collect four snapshots during the training process. Then, we average the outputs of these different snapshots for prediction. This approach also requires more inference time.

Besides, we conducted a full experimental validation of Logit Ensemble and SWA under a new dataset Stanford Cars, as shown in Table 9. Under the Stanford Cars dataset, Lookaround still performs better than Logit Ensemble. Additionally, adding SWA into the training process of Lookaround can further improve performance. Under the limit of single-model inference, Lookaround+SWA achieves a Top-1 accuracy of 91.04%, exceeding the optimal 88.11% (SWA with AutoAugment).

### C.5 Ablation Study

In our ablation experiments, we evaluate the individual contributions of two components: Data Augmentation (DA) and Weight Averaging (WA) to the effectiveness of the Lookaround. Below are the specific settings for different ablation experiments.

For the experiments without both DA and WA, three models were trained independently, each utilizing a distinct data augmentation strategy, for a duration of 200 epochs. After training, we identified

and reported the highest accuracy achieved by these models on the test set. In the experiments that employed only DA, we trained a single model using all data, combining the three data augmentation strategies, without the use of weight averaging. For the experiments with only WA, three models were trained separately for 200 epochs each, employing different data augmentation strategies. Following this, we selected the model that showcased the best performance on the test set.

### C.6 Additional Analysis

In the robustness experiments with the number of Data Augmentation methods, the six data augmentation methods are given as: RandomVerticalFlip, RandomHorizontalFlip, RandAugment, AutoAugment, RandomPerspective, RandomEqualize. All the Augmentation methods are from the PyTorch library. When more data augmentation methods are used, the training time will be correspondingly increased in our method. Therefore, in this paper's main experiments, we only select three data augmentation methods for comparison to reduce the time consumption.

## D Contemporaneous Work

After completing our work, we discovered similar work at CVPR 2023 within the same year. Inspired by similar motivations, the DART [20] method proposed by Samyak Jain et al. utilizes a similar training structure that includes different data augmentation training of multiple sub-models, with weight averaging occurring at intervals.

**Methodology.** DART starts weight averaging until the second half training process and adopts a sparser weight averaging strategy. Meanwhile, the DART scheme uses the ERM [15] strategy to initialize the model. In contrast, Lookaround maintains a consistent and dense weight averaging strategy throughout its training, making it more user-friendly in real-world applications.

**Theory.** DART proves that the average while training is more robust from the perspective of noise feature. On the other hand, Lookaround proves that it can get lower expected loss under the setting of quadratic noise function.

**Experiments.** DART validates its effectiveness in domain generalization. Lookaround is tested in the scenarios of finetuning and training from scratch.

In conclusion, although sharing a similar idea, two methods differ substantially in theoretical foundations and experimental approaches. Both methodologies are complementary and furnish significant insights for the broader research community.

And we compared Lookaround with DART with the same data augmentation, the results are shown in Table 10. Our results were better than DART, the reason behind this may come from the fact that we used more frequent "average steps" during training and started "average steps" earlier in the cycle. More frequent "average steps" are more conducive to the training of Lookaround, as referenced in the main body of the paper.

Table 10: Comparison with the DART method with ResNet18 on CIFAR10 and CIFAR100. The results of DART[†] are taken from reference [20].

| Method | CIFAR10 | CIFAR100 |
|---|---|---|
| Dart[†] | 97.14 | 82.89 |
| Lookaround | 97.22 | 83.16 |

## E Relationship with Model Soups

Model Soups [46] is a framework for finetuning a common pretrained model using different hyperparameters and then averaging the weights of different finetuned models to improve model performance and generalization. In the finetuning settings, we conducted an in-depth comparison between Lookaround and Model Soups, with the related results presented in Table 11. The "Update BN" suggestion originated from the reviewer's feedback. Given that we employed a CNN architecture, the averaging of means and variances in the batch normalization (BN) layers of different models within Model Soups might lead to distortions, failing to accurately represent the statistical characteristics of the true hidden layer inputs. To address this, we passed the averaged model through

Table 11: Comparisons between Lookaround, Model Soups on ResNet50. "M=18" represents 18 different hyperparameters. "M=3" represents that Lookaround is trained using 3 different data augmentation techniques. 18 different configurations come from the combinations of three data augmentations (RandomHorizontalFlip, RandAugment, AutoAugment), three initial learning rates: $\{0.01, 0.003, 0.001\}$ and label smooth or not. Uniform Soups$^+$ represents that we use the top-10 accuracy models for uniform soups (the accuracy of these models ranges from 83.91% to 84.64%).

| Method | Update BN | Top-1 Acc | Top-5 Acc |
|---|---|---|---|
| Uniform Soups | - | 82.93 | 97.44 |
| Uniform Soups | ✓ | 84.53 | 98.10 |
| Uniform Soups$^+$ | - | 84.60 | 97.81 |
| Uniform Soups$^+$ | ✓ | 85.57 | 98.04 |
| Greedy Soups | - | 85.52 | 98.02 |
| Greedy Soups | ✓ | 86.09 | 98.11 |
| Lookaround (M=3) | - | 85.20 | 97.15 |

the entire training set using forward propagation to individually update the BN layer's running mean and running var.

The experimental results indicate that the Lookaround method, employing only three types of data augmentation, still remains comparable in performance to the Model Soups method, which utilizes 18 types of data augmentation. This further attests to Lookaround's generalization capability even under limited data augmentation conditions. Additionally, we observed that the Lookaround method is applicable to tasks with random initialization, while the Model Soups, due to its prerequisite of pretrained weights, cannot be employed in such scenarios.

## F  Computational Complexity and Comparison

The Lookaround optimizer can be viewed as maintaining multiple backups of model weights during the optimization process. Each backup is only trained under its corresponding data augmentation strategy (forward and backward propagation) and updates the parameters using weight averaging after multiple iterations.

In general, if the time for an iteration is defined as $\Omega$, and the time to perform a weight average is $\omega$, given that the dataset contains $B$ batches, the standard time complexity of SGDM to complete an epoch is $O(B\Omega)$. The time complexity of Lookaround is $O(dB\Omega + \frac{B\omega}{d})$. Given that $\omega$ is much smaller than $\Omega$, this can be approximated as $O(d\Omega B)$.

In our experiments (corresponding to Table 1 to 5 in the paper), to establish fair comparisons in computational among different approaches, we adopt the same augmentations for the competitors. Specifically, within a single epoch, both the proposed Lookaround and the competitors undergo training on an identical $d$ times the data augmentations. With such a setup, we guarantee consistency in the data volume utilized by each method, thereby ensuring fair comparisons in terms of computation.

