# OpenReview forum: "Lookaround Optimizer: $k$ steps around, 1 step average"
_NeurIPS.cc/2023/Conference — NeurIPS 2023 poster_

### Official Review · Reviewer_3YPD · 2023-07-02

**Soundness:** 3 good
**Presentation:** 4 excellent
**Contribution:** 3 good
**Rating:** 8
**Confidence:** 4

**Summary:**

The paper presents Lookaround, a novel optimizer for weight average ensembling (WA). Unlike existing approaches that perform weight averaging post-training, Lookaround adopts a two-step process throughout the training period. In each iteration, the "around" step trains multiple networks simultaneously on transformed data using different augmentations, while the "average" step combines these networks to obtain an averaged network as the starting point for the next iteration.The approach demonstrates clear superiority over state-of-the-art methods in extensive experiments on CIFAR and ImageNet datasets using both traditional CNNs and Vision Transformers. The paper provides theoretical justification and commits to open science by making the code publicly available.

**Strengths:**

1. `Innovative Approach`: The introduction of Lookaround as a straightforward and effective optimizer for weight average ensembling brings a novel perspective to the field. The two-step process during training enhances network diversity and preserves weight locality, addressing the limitations of post-hoc weight averaging approaches.

2. `Theoretical Justification`: The paper offers strong theoretical support for the superiority of Lookaround through convergence analysis. Proposition 1 provides insights into the convergence variance of Lookaround in comparison to typical SGD and Lookahead optimizers. This rigorous theoretical analysis enhances the credibility and significance of the proposed method, demonstrating a solid foundation for its effectiveness and performance.

3. `Extensive Experimental Validation`: The authors conduct extensive experiments on popular benchmarks, including CIFAR and ImageNet, with both traditional CNNs and Vision Transformers (ViTs). The clear superiority of Lookaround over state-of-the-art methods on these datasets demonstrates its effectiveness and applicability.

4. `Commitment to Open Science`: The authors state their intention to make the code publicly available, fostering reproducibility and enabling further research in the field.

**Weaknesses:**

1. `Computational Complexity and Comparison`: The paper lacks a comprehensive discussion of the computational complexity introduced by the Lookaround optimizer. As Lookaround incorporates an additional around step during training, it is crucial to assess its computational requirements and compare them to existing optimization methods. A detailed analysis of the computational trade-offs, including runtime and memory usage, would provide a more comprehensive understanding of Lookaround's practical applicability and scalability.

2. `Limited Experiments on Datasets Beyond Image Classification`: The paper primarily focuses on evaluating Lookaround on image classification tasks using datasets like CIFAR and ImageNet. However, to establish the versatility and effectiveness of Lookaround, it is important to explore its performance on datasets beyond image classification. Conducting experiments on diverse tasks such as language modeling or segmentation/detection tasks would demonstrate the generalizability of Lookaround across various domains and provide a more comprehensive evaluation of its performance.

**Questions:**

1. With the Lookaround optimizer requiring k around steps and 1 average step, I am unclear about how the epoch and iteration are defined in this setting. Are they counted based on the average step or the around step? Could you provide clarification on how the epoch and iteration are structured in Lookaround?

2. Considering that each batch is augmented k times in Lookaround, I am curious to understand how this approach differs from simply applying `repeated augmentations`[A] to each batch. What distinguishes the effect of k augmentations and 1 step update in Lookaround versus performing each step update individually and then averaging all checkpoints? In other words, what advantages does Lookaround's approach provide in terms of diversity and ensemble performance compared to the alternative method of averaging checkpoints from individual updates?

[A] Hoffer E, Ben-Nun T, Hubara I, et al. Augment your batch: Improving generalization through instance repetition[C]//Proceedings of the IEEE/CVF Conference on Computer Vision and Pattern Recognition. 2020: 8129-8138.

**Limitations:**

As discussed in the limitation section "additional cost of network searching using different data augmentation, resulting in a longer training time proportional to the the number of trained networks", the running time is a important concern.

---

> ### Author Rebuttal · Authors · 2023-08-10
>
> Thanks for your encouraging words and constructive comments. We sincerely appreciate your time in reading the paper, and our point-to-point responses to your comments are given below.
>
> **Q1: How are the epoch and iteration defined in Lookaround's setting?**
>
> Under the standard training process, an epoch means the model fully traverses the origin dataset once, typically consisting of $N$ batches. However, in the Lookaround setting, an epoch represents that the model fully traverses the dataset $d$ times, equivalent to $d*N$ batches. This is because each batch of data undergoes $d$ different data augmentations and is trained by the model. Lookaround, $k$ steps around, 1 step average. One iteration means that each branch of Lookaround does an around step. We will modify it in the pseudo-code section of the paper to make it easier to understand.
>
> **Q2: Lookaround versus repeated augmentations.**
>
> Thanks for mentioning this wonderful work. While our method does share some similarities with it, there are also significant differences. Firstly, the motivations behind both approaches are distinct. The central idea of the paper is to use multiple data augmentations within a single batch to average gradients, hoping to achieve a more stable gradient representation. In contrast, the goal of Lookaround is to increase the diversity of the model in the loss landscape. It performs gradient descent separately under different data augmentations $k$ times, followed by an average once. From this perspective, "Augment your batch" can be seen as a special case of the Lookaround method where $k$ is set to 1.
>
> **Q3: Computational Complexity and Comparison.**
>
> The Lookaround optimizer can be viewed as maintaining multiple backups of model weights during the optimization process. Each backup is only trained under its corresponding data augmentation strategy (forward and backward propagation) and updates the parameters using weight averaging after multiple iterations.
>
> In general, if the time for an iteration is defined as $\Omega$, and the time to perform a weight average is $\omega$, given that the dataset contains $B$ batches, the standard time complexity of SGDM to complete an epoch is $O(B\Omega)$. The time complexity of Lookaround is $O(dB\Omega+\frac{B\omega}{d})$. Given that $\omega$ is much smaller than $\Omega$, this can be approximated as $O(dB \Omega)$.
>
> In our experiments (corresponding to Table 1 to 4 in the paper), to establish fair comparisons in computational among different approaches, we adopt the same augmentations for the competitors. Specifically, within a single epoch, both the proposed Lookaround and the competitors undergo training on an identical d times the data augmentations. With such a setup, we guarantee consistency in the data volume utilized by each method, thereby ensuring fair comparisons in terms of computation.
>
> The following is an example of the running time of different methods in our experiments. We run each method for 50 epochs and calculate the average execution time of ResNet50 for one epoch on a 3090 graphics card on CIFAR100. The results are shown in Table S1.
> Table S1: Average execution time of each method traversing an epoch on ResNet50 on CIFAR100.
> |     Method     | SGDM | Lookahead | AdamW | SAM  | Lookaround |
> | :------------: | :--: | :-------: | :---: | :--: | :--------: |
> | Execution Time ($s$) | 123  |    123    |  125  | 246  |    123     |
>
>   When considering memory usage, we can divide it into two parts: one part is generated by the model parameters, and the other part is produced by the intermediate results during the model training process, where the latter occupies the main part of the memory. In terms of memory usage for model parameters, the memory occupied by parameters in the Lookaround method is d times that of SGDM. As for the intermediate results produced by the computational graph during model training, since it only updates on the corresponding weight backup and does not increase with the number of data augmentation strategies, its memory usage is the same as conventional methods (such as SGDM).
>
> **Q4: Limited Experiments on Datasets Beyond Image Classification.**
>
> We really appreciate your suggestions. It's a great idea to apply Lookaround to broader areas including language modeling and segmentation/detection tasks. In this work, we focuses on visual classification. The broader application are left to our future work. Thank you again for your kind suggestions positive comment on our work.

---

> > ### Comment · Reviewer_3YPD · 2023-08-13
> > **Response to the author**
> >
> > I greatly appreciate your feedback and the clarification you've provided. I am content with the overall quality of the paper and would like to vote for a "Strong Accept" (8).

---

> > > ### Author Response · Authors · 2023-08-13
> > >
> > > We express our gratitude to the reviewer for your positive feedback. We appreciate your valuable comments and we will carefully revise the paper based on your comments.

---

### Official Review · Reviewer_V9QB · 2023-07-04

**Soundness:** 4 excellent
**Presentation:** 3 good
**Contribution:** 2 fair
**Rating:** 6
**Confidence:** 2

**Summary:**

This work provides a new optimizer, "Lookaround optimizer," which is built upon a previous proposed "Lookahead optimizer" [40]. By incorporating data augmentation, this work shows an improved convergence rate under low condition numbers. It also empirically shows some improvement in classification accuracy under several datasets.

**Strengths:**

This work proposes a new optimizer and shows consistent (despite little) improvement in various settings. The supplementary material shows a solid derivation.

**Weaknesses:**

1. The major concern lies in the limited improvement over the "Lookahead optimizer." The improvement in Figure 2 and Table 1-3 all seems to be subtle. Since the proposed methods see more augmented data in each epoch than others, it is questionable whether the improvement actually comes from these additionally augmented data. Also, the idea itself is almost identical to the Lookahead optimizer, so the novelty is rather limited.

2. Minor mistakes: In table 1, row CIFAR 10, column ResNext50 Top5 results, the best results happen in Lookahead (99.96), not the proposed method (99.95).

**Questions:**

What is the d (number of data augmentation) used in the experiment? I cannot find it in the context.

**Limitations:**

The author addresses that the data augmentation process consumes extra time, which is likely to be the main limitation.

---

> ### Author Rebuttal · Authors · 2023-08-10
>
> Thanks for your encouraging words and constructive comments. We sincerely appreciate your time in reading the paper, and our point-to-point responses to your comments are given below.
>
> **Q1：What is the d (number of data augmentation) used in the experiment?**
>
> In the experiments corresponding to Table 1, Table 2, and Table 4, we adopted a setting with $d=3$. The three data augmentation methods are RandomHorizontalFlip, RandomVerticalFlip, and RandAugment. This information can be found in lines 221-223 of the paper and also in the appendix C.
>
> **Q2: Whether the improvement in accuracy comes from these additionally augmented data.**
>
> We place great emphasis on experimental settings, ensuring that the improvements observed in the experiments are primarily attributed to our method and not just to data augmentation. Specifically, within a single epoch, both the proposed Lookaround and the competitors undergo training on an identical $d$ times the data augmentations. With such a setup, we guarantee consistency in the data volume utilized by each method, thereby ensuring fair comparisons in terms of computation.
>
> The table below presents the experimental results of various networks on the CIFAR100 dataset. In this context, H(data), V(data), and R(data) each represent specific data augmentation applied to the original data. H(data) + V(data) + R(data) signifies training for one epoch after combining these three types of data augmentations. Throughout the 200 epochs of experimentation, Lookaround, a variant of SGDM, demonstrated significant performance improvements.
>
> Table S1: Top-1 accuracy of different networks under CIFAR100. Here "H" denotes random horizontal flip data augmentation. "V" denotes random vertical flip data augmentation. "R" denotes RandAugment data augmentation.
> |       Per epoch data        | Method | Vgg19 | ResNeXt50 | ResNet50 | ResNet101 | ResNet152 | Execution Time | amount of data |
> | :-------------------------: | :---------------: | :---: | :-------: | :------: | :-------: | :-------: | :------------: | :------------: |
> | H(data) + V(data) + R(data) |       SGDM        | 73.84 |   79.10   |  79.61   |   79.91   |   80.16   |       3x       |       3x       |
> | H(data) + V(data) + R(data) |    Lookaround     | 74.29 |   81.14   |  81.60   |   81.97   |   82.22   |       3x       |       3x       |
>
> **C3: Minor mistakes.**
>
> Thank you very much for pointing out this issue. We will fix it in the revised version.

---

> > ### Comment · Reviewer_V9QB · 2023-08-18
> >
> > Thanks to the author for the detailed responses that alleviate my concern on whether the improvement in accuracy comes from these additionally augmented data. However, for me, the novelty of the idea seems rather limited as it is almost identical to the Lookahead optimizer. Thus, I prefer to remain the rating as "borderline accept."

---

> > > ### Author Response · Authors · 2023-08-18
> > > **Thanks for your reply!**
> > >
> > > Thank you for your feedback and the positive rating given to our work. We would like to provide further clarification regarding the differences between Lookaround and Lookahead as follows.
> > >
> > > Despite their similar names, Lookaround and Lookahead are fundamentally distinct in their spirts and methodologies. The Lookahead optimizer combines fast and slow weights along the training trajectory to identify a lower loss basin. Intuitively, the algorithm chooses a search direction by looking ahead at the sequence of “fast weights" generated by another optimizer. However, Lookaround adopts an iterative weight average strategy throughout the whole training period to constantly make the diversity-locality compromise. The terms "foward, back" in Lookahead title and "around, average" in Lookaround title succinctly encapsulate their distinctions in both spirts and approaches. Furthermore, our work theoretically demonstrate that the proposed Lookaround can converge to a lower expected loss and a smaller steady-state variance, which means that the model is more stable and has a better ability to resist noise. At the same time, Lookaround also shows certain advantages in terms of convergence speed. These findings are further supported by extensive and comprehensive experiments, which validate the superior performance of Lookaround compared to the previous Lookahead method. At the same time, we observe that Lookaround has smaller oscillation and higher accuracy on the convergence curve of the test set, which further demonstrates our theory.
> > >
> > > We sincerely look forward to your reevaluation of our work and would greatly appreciate it if you could raise your score to boost our chance of greater exposure within the community. Thank you very much!

---

> > > > ### Comment · Reviewer_V9QB · 2023-08-22
> > > >
> > > > After viewing the author's response and discussion with other reviewers, I now agree that novelty is not the main issue here, so I raise my score to "week accept."

---

### Official Review · Reviewer_d83u · 2023-07-04

**Soundness:** 1 poor
**Presentation:** 2 fair
**Contribution:** 2 fair
**Rating:** 5
**Confidence:** 3

**Summary:**

This paper introduces a new optimization algorithm named Lookaround, which draws inspiration from the recent achievements of weight averaging (WA) techniques in deep learning. The proposed Lookaround optimizer looks around nearby points by performing multiple gradient computations for a given training input using different data augmentations and averages them to obtain better generalizing solutions.

**Strengths:**

The theoretical analysis in this paper largely relies on Zhang et al. (2019), including noisy quadratic analysis and deterministic quadratic convergence. While it does not introduce a novel form of analysis, it is significant as it establishes a theoretical foundation within the existing framework.

For experiments, the baseline comprises commonly employed optimization techniques, such as SWA (Izmailov et al., 2018) and SAM (Foret et al., 2021), which are widely accepted as standard approaches in the field. In addition to convolutional neural networks of various scales, the authors also considered experiments on vision transformers.

**Weaknesses:**

Experimental issues:

* __There are no error bars.__
While the authors answered "yes" for "Error Bars", the paper only provide a single value across the tables and figures. It is unclear how many experiments were conducted to derive those values. It would be preferable if the authors also included averages accompanied by standard deviations to provide a more comprehensive representation of the experimental results.

* __The outcomes obtained from the ImageNet experiments appear to be strange.__
Despite the authors' efforts to demonstrate the scalability of the proposed algorithm, the Top-1 accuracy of 72.27% reported in Table 2 appears to be comparatively low. After reviewing Appendix C.1.2, it seems that the experimental setup follows the PyTorch convention (https://github.com/pytorch/examples/tree/main/imagenet), except for some additional augmentations. It is widely recognized that the ResNet-50 model typically achieves an accuracy of approximately 76% on the ImageNet dataset using this standard setup, which significantly differs from the 72.27% accuracy reported by the authors. It would be beneficial to provide clarification on the reasons behind the considerable performance drop observed in the ImageNet results.

Practical issues:

* __Excessive training costs incurred by the proposed algorithm.__
The Lookaround algorithm, as proposed, demands $d$ times the number of forward and backward passes for each optimization step. This is considerably higher compared to SAM (Foret et al., 2021), which only requires twice the number of passes. One could argue that the training epoch is effectively enlarged by a factor of $d$ and this is the actual reason for the performance improvements. It would be valuable to present results akin to Table 2 in Foret et al. (2021), that is, exploring the impact of increasing the number of training epochs through an ablation study.

**Questions:**

There are several issues in the experimental results of the current version of the paper as I mentioned above, and I would respectfully suggest the followings that could make the paper solid:

* __The absence of SAM in Table 4.__
Is there a specific reason for excluding SAM from the results presented in Table 4?

* __The disparity in results for ResNet50 on CIFAR100 between Tables 1 and 6.__
Table 1 presents Top-1/5 accuracies of 81.60/95.99, while these values are not reflected in Table 6.

* __Top-5 accuracy does not provide a meaningful comparison in the main tables.__
It is suggested to exclude the Top-5 accuracy metric from the main results presented in Tables 1 and 4 for CIFAR10/100.  Specifically, in the case of CIFAR-10, evaluation metrics where all techniques achieve 99.9% accuracy hold little significance. Instead, including the Top-5 accuracy in Table 2 would be more appropriate, as it provides a more meaningful performance evaluation for the ImageNet experiments.

* __It would be nice to provide uncertainty metrics.__
While the Top-1 accuracy is the main evaluation metric for classification, it would be advantageous to incorporate additional metrics that can effectively demonstrate the superiority of the proposed algorithm. For instance, including the negative log-likelihood (NLL) as a measure of in-domain uncertainty would provide valuable insights into the algorithm's performance.

* __It would be beneficial to offer an analysis of the training costs involved.__
Several algorithms, including the proposed one, impose additional computational burdens, which could impact their practicality (e.g., SAM requires performing double forward and backward passes; SWA requires an additional model copy in memory). Providing detailed information about the specific computational requirements of each algorithm would greatly benefit future researchers and engineers. The most straightforward analysis is measuring wall-clock time for training. It would be nice to replace the meaningless "#param." column with the training runtime in Table 4.

**Limitations:**

The authors stated the additional training cost incurred by the proposed algorithm.

---
__References:__
Zhang et al. (2019). _Lookahead optimizer: k steps forward, 1 step back_.
Izmailov et al. (2018). _Averaging weights leads to wider optima and better generalization_.
Foret et al. (2021). _Sharpness-aware minimization for efficiently improving generalization_.

---

> ### Author Rebuttal · Authors · 2023-08-10
>
> Thank you for your constructive comments. In the following, your comments are first stated and then followed by our point-by-point responses.
>
> **Q1: There are no error bars.**
>
> The error bar (i.e., standard deviations) is depicted in Figure 6 in our paper. The detailed results are provided in Table S1. We will make it clearer in the revised version.
>
> Table S1: Top-1 accuracy of Lookaround (with standard deviation).
> |   Method   | Dataset|VGG19|Resnet50|Resnet101|Resnet152|ResNext50|
> | :--------: | :------: | :---------------: | :---------------: | :---------------: | :---------------: | :---------------: |
> | Lookaround | CIFAR10  | 94.42 ($\pm$0.09) | 96.56 ($\pm$0.02) | 96.77 ($\pm$0.08) | 96.94 ($\pm$0.12) | 96.67 ($\pm$0.11) |
> | Lookaround | CIFAR100 | 74.2 ($\pm$0.25)  | 81.12 ($\pm$0.63) | 81.89 ($\pm$0.53) | 81.99 ($\pm$0.24) | 81.17 ($\pm$0.51) |
>
> **Q2: The outcomes obtained from the ImageNet experiments appear to be strange.**
>
> For the sake of fair comparisons in training data and computation cost, each method (including Lookaround and the competitors) was trained with the same data augmentations (random horizontal flip, random vertical flip, and RandAugment). With such a setup, we guarantee consistency in the data volume utilized by each method, thereby ensuring fair comparisons in terms of computation. The divergence in performance observed between Lookahead, as reported in the original paper and the current study, can be attributed to the differing data augmentation techniques employed.
>
> **Q3: Excessive training costs incurred by the proposed algorithm.**
>
> In order to compare the performance of Lookaround and standard SGDM training more fairly, we shorten the number of epochs of Lookaround training to 1/4 of SGDM. As shown in Table S2, Lookaround's performance is still competitive.
>
> Table S2: Comparison between SGDM and Lookaround with ResNet50 on CIFAR100. Here "H" denotes random horizontal flip data augmentation. "V" denotes random vertical flip data augmentation. "R" denotes RandAugment data augmentation.
> |    Method   | Data Augmentation Strategy | Epoch | CIFAR10 | CIFAR100 | Training time |
> | :--------: | :------------------------: | :---: | :-----: | :------: | :-----------: |
> |SGDM|"R"|200|95.57|78.59|1 Budget|
> |SGDM|"H" + "V" + "R"|200|95.96 |79.61|3 Budget|
> | Lookaround |"H" + "V" + "R"|50|95.82 |78.62|3/4 Budget|
> | Lookaround |"H" + "V" + "R"|200|**96.59** |**81.60**|3 Budget|
>
> **Q4: The absence of SAM in Table 4.**
>
> We supplement the SAM experiments of ResNet50 and ViT-B/16 under pre-training on Table S3. In the pre-training experiment, Lookaround and SAM significantly improve the standard training process, among which SAM is more suitable for ViT architecture, while Lookaround is more suitable for CNN architecture. At the same time, the two are orthogonal, and combining the two can achieve higher performance improvement.
>
> Table S3: Top-1 accuracy of SAM and Lookaround on CIFAR dataset under pre-training.
> |Backbone |Method|  CIFAR10  | CIFAR100  |
> |:------: | :------------: | :-------: | :-------: |
> |ResNet50 |SGDM|96.08|82.04|
> |ResNet50 |SAM|97.02|82.75|
> |ResNet50 |Lookaround|96.79|83.62|
> |ResNet50 |Lookaround+SAM|**97.53**|**83.91**|
> |ViT-B/16 |Adam|92.91|74.50|
> |ViT-B/16 |SAM|98.02|89.13|
> |ViT-B/16 |Lookaround|95.23|78.38|
> |ViT-B/16 |Lookaround+SAM|**98.84**|**92.04**|
>
>
> **Q5: The disparity in results for ResNet50 on CIFAR100 between Tables 1 and 6.**
>
> The result differences of ResNet50 on CIFAR100 in Table 1 and Table 6 come from the different settings of $k$. In Table 1, the choice of $k$ is 5, meaning an average is taken every 5 batches. In Table 6, the parameter $k$ we chose is "one-epoch", indicating that in Lookaround, different branches train for a complete epoch each before performing a weight averaging. Thank you very much for pointing this out, we supplement this experiment in Table S4.
>
> Table S4: Top-1 accuracy (%) of different data augmenta-
> tion (DA) number by using ResNet50 on CIFAR100 dataset.
> |# of DA|1|2|3|4|5|6|
> |:-------:|:--:|:--:|:--:|:--:|:--:|:--:|
> |Top-1 (%)|78.2|80.82|81.60|81.19|81.74|82.02|
> |Top-5 (%)|94.5|95.19|95.99|95.65|95.85|96.02|
>
> **Q6, Q7: Top-5 accuracy does not provide a meaningful comparison in the main tables. It would be nice to provide uncertainty metrics.**
>
> Thank you very much for your suggestion. We will revise the paper accordingly following your suggestions. We present the NLL loss for the test set for part of the experiment, and the data in this table correspond to the CIFAR100 dataset in Table 1 in the paper.
>
> Table S5: NLL Loss values of different methods under ResNet family.
> |Method|ResNet50|ResNet101|ResNet152|ResNext50|
> |:--------:|:-------:|:-------:|:-------:|:-------:|
> |SGDM|0.899|0.894|0.895|0.820|
> |SWA|0.858|0.856|0.856|0.796|
> |Lookahead|0.836|0.825|**0.809**|0.787|
> |Lookaround|**0.81**|**0.823**|0.823|**0.73**|
>
> **Q8: It would be beneficial to offer an analysis of the training costs involved.**
>
> In the experiment of comparing Lookaround with other optimization methods, we train each method with the same number of epochs, and the corresponding data of each epoch comes from the sum of $d$ times of augmentation data. Therefore, non-SAM methods will have the same number of forward and back propagation, while SAM requires twice as many forward and back propagation. Although each optimizer will update momentum or average weight, the time consumption of this step in the overall training is very small.
>
> Experimentally, we run each method for 50 epochs and calculate the average execution time of ResNet50 for one epoch on a 3090 graphics card on CIFAR100. The results are shown in Table S6.
>
> Table S6: Average execution time of each method traversing an epoch on ResNet50 on CIFAR100.
> |Method| SGDM | Lookahead | AdamW | SAM  | Lookaround |
> | :------------: | :--: | :-------: | :---: | :--: | :--------: |
> | Execution Time ($s$)|123|123|125|246|123|

---

> > ### Comment · Reviewer_d83u · 2023-08-11
> >
> > Thank you for the author's effort.
> >
> > > __Q2: The outcomes obtained from the ImageNet experiments appear to be strange.__
> > >
> > > For the sake of fair comparisons in training data and computation cost, each method (including Lookaround and the competitors) was trained with the same data augmentations (random horizontal flip, random vertical flip, and RandAugment). With such a setup, we guarantee consistency in the data volume utilized by each method, thereby ensuring fair comparisons in terms of computation. The divergence in performance observed between Lookahead, as reported in the original paper and the current study, can be attributed to the differing data augmentation techniques employed.
> >
> > The checkpoint `ResNet50_Weights.IMAGENET1K_V1` from torchvision indicates a 76.13% accuracy using basic training receipt involving the SGD optimizer. However, the performance detailed in the current manuscript is significantly below expectations, despite using nearly identical settings (90 training epochs; MultiStepLR scheduler with decay steps at 30th and 60th epochs and a decay factor of 0.1; initial learning rate of 0.1 and a batch size of 256). This substantial difference in results might arise from flaws in the experimental configuration, raising concerns about the reliability of the experimental outcomes presented in the paper. Consequently, additional clarifications of why it happens are imperative to address this issue.

---

> > > ### Author Response · Authors · 2023-08-12
> > >
> > > Dear Reviewer,
> > >
> > > we sincerely appreciate your prompt feedback on our rebuttal, and we deeply apologize for the delay in our response, which was due to the extensive clarification experiments conducted on Imagenet. We would now like to provide further clarification as follows.
> > >
> > > As we have previously mentioned in our rebuttal, in order to ensure fair comparisons in terms of computation, we have employed the same multi-augmentation techniques for all methods. Additionally, we have incorporated a one-epoch warm-up phase, as detailed in Appendix C.1.2, for evaluating all methods in our experiments. This warm-up phase has proven effective in all our CIFAR experiments, and thus, we maintained its inclusion in the ImageNet experiments. However, after conducting extensive and rigorous experiments, we discovered that the warm-up phase did not yield optimal results when combined with multi-augmentation on ImageNet. Detailed results can be found in Table S7. Consequently, by removing the warm-up phase, all our experiments on the competitors achieved performance comparable to the results published in prior works. Notably, the proposed Lookaround once again outperformed these baseline methods.
> > >
> > > We sincerely apologize for any confusion caused by our experimental results. We firmly believe that a superior optimizer should consistently yield improved results across various settings, rather than being limited to specific settings. The proposed Lookaround demonstrates superior performance in both our original settings (with warm-up) and the revised settings (without warm-up), further confirming the efficacy of our method.
> > >
> > > We will updata these results in the revised version. Thank you for your understanding and consideration.
> > >
> > > Best regards,
> > > The authors of Lookaround
> > >
> > > Table S7: Comparison between SGDM and Lookaround with ResNet50 on ImageNet. Here "H" denotes random horizontal flip data augmentation. "V" denotes random vertical flip data augmentation. "R" denotes RandAugment data augmentation.
> > > |    Method   | Data Augmentation Strategy  | Warm Up | Top-1(%) | Top-5(%) |
> > > | :--------: | :------------------------: | :------: | :------: | :------: |
> > > |SGDM|"H" + "V" + "R"| &#x2713; |72.27 |90.99|
> > > |Lookaround |"H" + "V" + "R"| &#x2713; |75.11|92.43|
> > > |SGDM|"H" + "V" + "R"| &#x2717; |75.97 |92.89|
> > > |**Lookaround** |"H" + "V" + "R"| &#x2717; |**77.32**|**93.29**|

---

> > > ### Author Response · Authors · 2023-08-19
> > >
> > > Dear reviewer,
> > >
> > > We sincerely thank you for your constructive comments.
> > >
> > > As suggested, we have conducted comprehensive experiments on ImageNet, comparing the performance of SWA, SGDM, Lookahead, and Lookaround. The results, as presented in Table S8, again demonstrate that Lookaround achieves notable performance improvements when no warm-up setting is employed.
> > >
> > > Table S8: Comparison between SGDM, SWA, Lookahead and Lookaround with ResNet50 on ImageNet. Results of SWA$^+$, Lookahead$^+$ are taken from the origin paper.
> > > |    Method     | Warm Up | Top-1(%) | Top-5(%) |
> > > | :--------:  | :------: | :------: | :------: |
> > > |SGDM| &#x2717; |75.97 |92.89|
> > > |SWA| &#x2717; |76.78 |93.18|
> > > |SWA$^+$| &#x2717; |76.97 |-|
> > > |Lookahead| &#x2717; |76.52 |93.11|
> > > |Lookahead$^+$| &#x2717; |75.49 |-|
> > > |**Lookaround** | &#x2717; |**77.32**|**93.29**|
> > >
> > > The allocated time for reviewer-author discussion is nearing its end, we sincerely look forward to your reevaluation of our work and would greatly appreciate it if you could raise your score to boost our chances of gaining more exposure in the community. Thank you very much!
> > >
> > > Best regards,
> > > The authors of Lookaround

---

> > > > ### Comment · Reviewer_d83u · 2023-08-19
> > > >
> > > > I appreciate all the efforts from the authors.
> > > >
> > > > The outcomes displayed in Table S7, which demonstrate a performance decrease in ResNet50 on ImageNet due to the warm-up phase, are somewhat puzzling in consideration of my own experiences or previous research findings (like He et al. 2019). Nonetheless, I will withhold my assessment of this aspect since the influence of RandAugment could have introduced some intricate factors. I believe newly attained ImageNet results that align with widely recognized numbers will make the paper solid and reliable.
> > > >
> > > > My main concerns (including previous issues with ImageNet performance) are that the results of the large-scale experiments presented in this paper show somewhat significant differences from previously reported values. I believe these new optimizer proposals should showcase overall benefits (if not all) in widely accepted experimental benchmarks, all the while being consistent with results from previous literature. For instance, Foret et al. (2021) clearly demonstrated the superiority of their proposed optimizer compared to previously reported state-of-the-art numbers (Table 3).
> > > >
> > > > However, the current manuscript does not seem to achieve this, and thus I would like to maintain my initial assessment. As a concrete example for ViT-B/16 experiments, the following table summarizes comparison with numbers from previous literature, where there are immoderate differences in performance:
> > > >
> > > > | Method | ViT-B/16 on CIFAR10 | ViT-B/16 on CIFAR100 |
> > > > | :- | :- | :- |
> > > > | Adam (pre-trained on ImageNet-21k; this work) | 92.91 | 74.50 |
> > > > | SWA (pre-trained on ImageNet-21k; this work) | 93.27 | 76.05 |
> > > > | Lookahead (pre-trained on ImageNet-21k; this work) | 94.21 | 77.57 |
> > > > | Lookaround (pre-trained on ImageNet-21k; this work) | 95.23 | 78.38 |
> > > > ||
> > > > | Adam (pre-trained on ImageNet; Dosovitskiy et al., 2021) | 98.13 | 87.13 |
> > > > | Adam (pre-trained on ImageNet-21k; Dosovitskiy et al., 2021) | 98.95 | 91.67 |
> > > > | Adam (pre-trained on JFT-300M; Dosovitskiy et al., 2021) | 99.00 | 91.87 |
> > > > ||
> > > > | Adam (pre-trained on ImageNet; Chen et al., 2022) | 98.1 | 87.6 |
> > > >
> > > > ---
> > > > (He et al., 2019) Bag of Tricks for Image Classification with Convolutional Neural Networks.
> > > > (Foret et al., 2021) Sharpness-Aware Minimization for Efficiently Improving Generalization.
> > > > (Dosovitskiy et al., 2021) An Image is Worth 16x16 Words: Transformers for Image Recognition at Scale.
> > > > (Chen et al., 2022) When Vision Transformers Outperform ResNets without Pre-training or Strong Data Augmentations.

---

> > > > > ### Author Response · Authors · 2023-08-20
> > > > >
> > > > > Thanks for your constructive comments. Before your comments, reviewer vFuf had also raised similar concerns.
> > > > >
> > > > > > Quality concerns regarding the baselines (both ResNet and ViT) in Table 4.
> > > > >
> > > > > We apologize that, in the early stages of the experiment, we focused too much on comparing different methods and overlooked the importance of establishing a good baseline for the pre-training process. Therefore, we restarted the pre-training experiments based on the settings in [1] and [2], and the results are shown in Table S9.
> > > > >
> > > > > One detail that needs to be pointed out is that the pretrained weights of the ResNet and ViT networks used in our paper were sourced from ImageNet-1k, not ImageNet-21k. This contradicts the description of our actual settings in your table.
> > > > >
> > > > > In our comparison experiments with the new baselines, we found that the results of ViT-B/16 under different optimizers were better than those reported in the original paper [2]. This is mainly due to our use of more data augmentation during the training process, aiming for a fair comparison with Lookaround. This result also indicates that Lookaround can still achieve better performance under improved baseline conditions.
> > > > >
> > > > > Thank you once again for your ongoing responses and concerns during the rebuttal phase. By comparing against the new baselines, we believe our Lookaround method will be more reliable.
> > > > >
> > > > > Table S9: The test set accuracy under the training procedure with pre-trained models using ImageNet-1k weights. The results of ViT-B/16$^+$ are taken from [2].
> > > > > | Backbone |   Method   | CIFAR10 | CIFAR100 |
> > > > > | :------: | :--------: | :-----: | :------: |
> > > > > | ResNet50 |    SGDM    |  97.55  |  84.50   |
> > > > > | ResNet50 |    SWA     |  97.48  |  84.81   |
> > > > > | ResNet50 | Lookahead  |  97.65  |  84.78   |
> > > > > | ResNet50 | **Lookaround** |  **97.82**  |  **85.20**   |
> > > > > | ViT-B/16 |    Adam    |  98.34  |  91.55   |
> > > > > | ViT-B/16$^+$ |    Adam    |  98.13  |  87.13   |
> > > > > | ViT-B/16 |    SWA     |  98.47  |  91.32   |
> > > > > | ViT-B/16 | Lookahead  |  98.51  |  91.76   |
> > > > > | ViT-B/16 | **Lookaround** |  **98.71**  |  **92.21**   |
> > > > >
> > > > > [1] When Vision Transformers Outperform ResNets without Pre-training or Strong Data Augmentations.
> > > > > [2] An Image is Worth 16x16 Words: Transformers for Image Recognition at Scale.

---

> > > > > > ### Comment · Reviewer_d83u · 2023-08-20
> > > > > >
> > > > > > Apologies for the misunderstanding. I misinterpreted the configuration in the supplementary code initially. I have now verified that `B_16_imagenet1k` is being utilized.
> > > > > >
> > > > > > I believe that the experimental outcomes revised during the rebuttal period would enhance the reliability of the experimental results and thus make the paper solid. Aligning the results with other studies (not necessarily flawless, but to a certain extent) holds importance not just in establishing the paper's credibility but also in preventing confusion among future researchers. I trust the authors comprehend this, even though redoing the experiment might have been somewhat cumbersome. I appreciate the authors again for all their efforts.
> > > > > >
> > > > > > Expecting that the comments showcased during the rebuttal phase will be systematically arranged in the final manuscript, I raise my rating above the borderline.

---

### Official Review · Reviewer_vFuf · 2023-07-06

**Soundness:** 3 good
**Presentation:** 2 fair
**Contribution:** 3 good
**Rating:** 5
**Confidence:** 4

**Summary:**

This paper proposes Lookaround, a new optimization method that incorporates weight averaging into the optimization process. The algorithm consists of two steps: 1) the around step launches several parallel runs of gradient descent led by different data augmentations, 2) the average step does weight averaging of the networks obtained by these parallel runs. These two steps are repeated along the whole training process, with the result of the average step being a starting point for the next around step. The proposed method is similar to Stochastic Weight Averaging and Model Soups, though it does the averaging during training rather than at the end of training. Lookaround shows strong results compared to the existing optimization methods (SGD with momentum, AdamW, Lookahead, SAM).

**Strengths:**

1. The idea of using weight averaging during optimization with various augmentations is novel and leads to a better generalization of neural networks.
2. This paper has a theoretical analysis of the quadratic noise setup which demonstrates faster convergence of Lookaround compared to SGD and Lookahead.

**Weaknesses:**

Despite the fact that the paper proposes a novel and interesting method, in my opinion, it is not quite ready for publishing in its current form. I believe that addressing the following concerns could help to make it a much stronger submission.
1. The text of the paper is hard to follow, contains some vague parts and numerous typos.
    - The theoretical reasoning is heavily inspired by the Lookahead paper, which makes it impossible to understand the theory without reading the original paper (e.g., I could not understand Proposition 1 and the definition of $\alpha$ without reading the Lookahead paper). Moreover, there is no remark that equations 4 and 5 are derived in the Lookahead paper.
    - Line 5-6: weight averaging and ensembles are mixed up.
    - Line 313: it is not clear how gradient boosting is related to this setup.
2. I have found two contemporaneous works published on arxiv (https://arxiv.org/abs/2302.14685, https://arxiv.org/abs/2304.03094) that propose approaches very similar to Lookaround. I believe they need to be at least discussed in the related work section.
3. The experimental part of the paper raises some questions and is not convincing enough, in my opinion.
    - The main results (Table 1) lack standard deviations, even though they are claimed in the OpenReview form.
    - The baselines seem too weak, i.e., the Lookahead paper reports >75% ImageNet test accuracy on ResNet-50 for SGD baseline, while this paper shows 72.27%.
    - The augmentation policy during baseline training is not clear. If the same set of augmentations as for Lookaround is used, then this may be the reason for the bad quality of the baselines.
    - Despite the paper proposing a new optimization method, it lacks experiments on a wider choice of architectures. Table 1 illustrates only VGG-19 and four networks from the ResNet family. It would be beneficial to add experiments on more modern convolutional architectures and more extensive experiments on image transformers.
    - The paper lacks a fair comparison to logit ensembles. If I understood correctly, different models of the ensemble are trained with different Lookaround augmentations. This leads to the suboptimal quality of each model and, as a result, to a suboptimal ensemble.
    - The comparison to model soups in Appendix D is not fair as well, i. e., the original paper fine-tunes models with various hyperparameters and does a greedy search for the combination with the best validation accuracy. Near zero accuracy of the average of $\theta_1$ and $\theta_2$ may be due to improper fine-tuning hyperparameters. Moreover, this is an important baseline, and it would be better to move this comparison to the main part of the paper.
    - A proper ablation study of augmentations is required. Is it possible to train Lookaround without augmentations for the around step “branches” (e.g., the only source of randomness is batch ordering)? What if each branch utilizes the same set of augmentations? Why are the mentioned augmentations used (and not some other ones, e. g. vertical flip seems to be a strange augmentation)? Probably, some of these setups are covered in Section 4.4, but it is not clear from the text.
    - Top-5 accuracy is redundant and makes it difficult to read the tables.
4. Minor issues/typos:
    - Line 25: the sentence is about LMC, but the citation [7] leads to the paper about permutations (Entezari et al., 2022).
    - Line 112-113: we provide
    - Line 147-148: repetition
    - Line 156: we analyze
    - Line 162: it seems that it should be $\mathbb{E} [c_i^2]$
    - Line 242: CIFAR

**Questions:**

1. What is the augmentation policy during baseline training?
2. I do not understand the experimental setup in the ablation study (Section 4.4). Is the training without DA and WA similar to the regular training without augmentations? Is the training without DA but with WA similar to Lookaround, where the ordering of batches is the only source of randomness between branches?
3. Section 3.2.2, convergence on deterministic quadratic functions. How branches of Lookaround are different from each other if there is no randomness in gradient steps?
4. Could the authors elaborate on how the Lookaround method relates to the methods proposed in the contemporaneous works (https://arxiv.org/abs/2302.14685, https://arxiv.org/abs/2304.03094)?

**Limitations:**

The main limitation of the proposed method is the increased training budget, which is highlighted by the authors. However, there is no fair comparison to networks trained for a larger number of epochs.

---

> ### Author Rebuttal · Authors · 2023-08-10
>
> Thank you for your detailed comments. We hope the following response will address your concerns.
>
> **C1: Two contemporaneous works propose approaches very similar to Lookaround.**
>
> Thanks for sharing the two great works! After carefully reading the two papers, we found these three works share moderate similarities, meanwhile with some notable differences:
>
> **Methodolegy.** DART starts weight averaging until the second half training process. PAPA do weight averaging more frequently like Lookaround throughout the training process. However, PAPA combines the weight of the sub-model with the average weight. The weighting factor varies in different training stages. Such a strategy is more sophisticated, yet introduces more hyperparameters, which increases the complexity of the method. Lookaround adopts a consistent weight averaging strategy throughout the training process, making it easier to use in practice.
>
> **Theory.** DART proves that the average while training is more robust from the perspective of noise feature. Lookaround proves that it can get lower expected loss under the setting of quadratic noise function. PAPA mainly focuses on empirical validation, without any theoretical analysis.
>
> **Experiments.** PAPA mainly makes comparisons with Model Soups. DART validate its effectiveness in domain generalization. Lookaround is tested in the senarios of finetuning and training from scratch.
>
> In conclusion, although sharing a similar essence, three methods differ substantially in theoretical foundations, experimental approaches, and method instantiations. They complement one another and each offers noteworthy contributions to the research community.
>
> **C2: The experimental part of the paper raises some questions.**
>
> **The standard deviations**. The error bar (i.e., standard deviations) is depicted in Figure 6 in our paper. The error bar of Lookaround trained from scratch are provided in Table S1. We will make it clearer in the revised version.
>
> Table S1: Top-1 accuracy of Lookaround (with standard deviation).
> |Method|Dataset|VGG19|Resnet50|Resnet101|Resnet152|ResNext50|
> |:-:|:-:|:-:|:-:|:-:|:-:|:-:|
> |Lookaround|CIFAR10|94.42 ($\pm$0.09)|96.56 ($\pm$0.02)|96.77 ($\pm$0.08)|96.94 ($\pm$0.12)|96.67 ($\pm$0.11)|
> |Lookaround|CIFAR100|74.2 ($\pm$0.25)|81.12 ($\pm$0.63)|81.89 ($\pm$0.53)|81.99 ($\pm$0.24)|81.17 ($\pm$0.51)|
>
> **The augmentation policy and the ImageNet accuracy**: For the sake of fair comparisons in training data and computation cost, each method (Lookaround and its competitors) was trained with the same data augmentations (random horizontal flip, random vertical flip, and RandAugment). With such a setup, we guarantee consistency in the data volume utilized by each method, thereby ensuring fair comparisons in terms of computation. The divergence in performance of Lookahead can be attributed to the differing data augmentation techniques employed.
>
> **More modern architectures**: We have tested the proposed Lookaround optimizer with diverse and popular model architectures including VGG19, ResNet, ResNext50 and ViT-B/16. We believe these representative achitecture are sufficient to validate the effectiveness of the proposed method across architectures.
>
> **Fair comparison to logit ensembles:** We have tried different training strategies for the base models in Logit Ensemble, including training different models with different augmentations, and training all the base models with the same mixed augmentations. In our experiments, no matter how the base models are trained, Lookaround exhibit consistently superior performance to logit ensembles. We will revise the paper to make this point clearer.
>
> **The comparison to model soups:** Here we would like to remind the reviewer that Lookaround is proposed for a quite different problem setting compared to Model Soups. Model Soups is designed for more efficient utilization of existing models, assuming that these models have already been pre-trained on specific datasets. Lookaround, on the other hand, serves as an optimizer for training deep models, devoid of any assumptions regarding the initialization of the model parameters. This distinction is why we included the comparison with Model Soups in the supplementary material. In terms of fairness, we have conducted experiments using both greedy search and uniform search. The results are presented in Table S2, where Lookaround outperforms the greedy Model Soups, further validating the effectiveness of our approach.
>
> Table S2: Comparison between Lookaround and Model Soups. "M=18" represents 18 different hyperparameters in Model Soups. "M=3" represents 3 different data augmentation techniques.
> |Optimizer|Method (M=18)|CIFAR100|
> |:-:|:-:|:-:|
> |SGDM|Model Soups (Greedy)|82.07|
> |SGDM|Model Soups (Uniform)|1.00|
> |AdamW|Model Soups (Greedy)|79.18|
> |AdamW|Model Soups (Uniform)|78.28|
> |**Lookaround**|Lookaround (M=3)|**83.62**|
>
> **Ablation study**: In Section 4.4 of the paper we set up the exact experiment you're talking about. The only WA experiment represents that each branch of Lookaround utilizes the same set of augmentation. We use the random vertical flip method as it can bring more image diversity, which is conducive to the guide model to have the characteristics of rotation invariance. We will specify these differences in the revised version.
>
> **C3: How branches of Lookaround are different from each other if there is no randomness in gradient steps?**
>
> This issue can be referred to in Appendix B.2. Here, we have made an approximate calculation. The idea behind Lookaround is to choose models from different branching points in the loss landscape for weight averaging. In this deterministic quadratic function, we chose model points corresponding to every point in the $k$-step trajectory for weight averaging.
>
> **C4: The text of the paper is hard to follow.**
>
> We are sorry for any confusion caused by our writing. We will fixed the issues you mentioned in our revisions.

---

> ### Author Response · Authors · 2023-08-15
>
> Dear reviewer,
>
> We sincerely appreciate the questions you posed during the initial rebuttal phase. In response to your first rebuttal, we have clarified that our experiments were conducted with strict fairness and will provide a more detailed explanation of the theory in the revised version, addressing your concerns with care. As we near the midpoint of the author-reviewer discussion stage, we look forward to receiving more feedback and engaging in productive conversations with you.
>
> Best regards,
> The authors of Lookaround

---

> > ### Comment · Reviewer_vFuf · 2023-08-15
> >
> > Thank you for the response and additional clarifications.
> >
> > Could you please elaborate more on the following points:
> > 1. **ImageNet experiments.** Could you please provide results for the baselines other than SGDM for the ImageNet experiments without warmup?
> > 2. **Model Soups.** Your results for model soups do not match the original model soups paper and the DiWA paper [1] (and also my own experience): uniform model soups should work much better than random. Could you please explain what are the reasons for such different behavior in your opinion? I suspect the reason is the poor choice of the fine-tuning procedure, which goes too far from the pre-trained checkpoint. Moreover, this procedure seems to lose the benefits of pre-training since it shows much worse results than the ones in the literature (84.5% in [2] and 86.4% in [3] v.s. yours 82%). Also, did you update batch norm parameters for model soups?
> > 3. **Logit Ensembles.** Your results for logit ensembles are quite surprising, hence, I am not fully convinced by the experiments on one dataset. Did you make similar experiments on other datasets? Did you try to compare Lookaround to an ensemble of SWA models?
> >
> > [1] Ramé et. al. Diverse Weight Averaging for Out-of-Distribution Generalization. NeurIPS 2022. \
> > [2] Kornblith at. al. Do Better ImageNet Models Transfer Better? CVPR 2019. \
> > [3] Grill et. al. Bootstrap Your Own Latent A New Approach to Self-Supervised Learning. NeurIPS 2020.

---

> > > ### Author Response · Authors · 2023-08-18
> > > **Reply to the second comment [1/2]**
> > >
> > > We appreciate your feedback on our rebuttal. Please find our responses to your questions below.
> > >
> > > > Could you please provide results for the baselines other than SGDM for the ImageNet experiments without warmup?
> > >
> > > Thanks for your comment. We have conducted comprehensive experiments on ImageNet, comparing the performance of SWA, SGDM, Lookahead, and Lookaround. The results, as presented in Table S3, again demonstrate that Lookaround achieves notable performance improvements when no warm-up setting is employed.
> > >
> > > Table S3: Comparison between SGDM, SWA, Lookahead and Lookaround with ResNet50 on ImageNet. Results of SWA$^+$, Lookahead$^+$ are taken from the origin paper.
> > > |    Method     | Warm Up | Top-1(%) | Top-5(%) |
> > > | :--------:  | :------: | :------: | :------: |
> > > |SGDM| &#x2717; |75.97 |92.89|
> > > |SWA| &#x2717; |76.78 |93.18|
> > > |SWA$^+$| &#x2717; |76.97 |-|
> > > |Lookahead| &#x2717; |76.52 |93.11|
> > > |Lookahead$^+$| &#x2717; |75.49 |-|
> > > |**Lookaround** | &#x2717; |**77.32**|**93.29**|
> > >
> > > >  Your results for model soups do not match the original model soups paper and the DiWA paper [1] (and also my own experience): uniform model soups should work much better than random.
> > >
> > > Here we would like to remind the reviewer that uniform soups do not always  improve model performance, which is also acknowledged in the original Model Soups paper (Please refer to Table J.1 in the original paper). It is mentioned in Appendix J.1 of Model Soups [1] that the learning rate lower than 1e-4 is easy to lead to the failure of weight average. This can be a reason for the failure of uniform soups under SGDM and the success of AdamW. However, as an optimizer that is sensitive to learning rates, SGDM's fine tuning performance is very poor at low learning rates. Thus we don't get a well-behaved uniform soups.
> > >
> > > > Could you please explain what are the reasons for such different behavior in your opinion? I suspect the reason is the poor choice of the fine-tuning procedure, which goes too far from the pre-trained checkpoint.
> > >
> > > The success of uniform soups in Model Soups heavily relies on stringent conditions. Specifically, the individual sub-models within uniform soups must remain within the same low-loss basin to ensure effective weight averaging and maintain high accuracy. However, when an excessive number of models is utilized, and in the presence of varying hyperparameter configurations, the use of data augmentation as a strong hyperparameter change can easily cause certain models to deviate and enter different low-loss basins, resulting in the failure of weight averaging. Consequently, achieving success with uniform soups requires careful adjustment of hyperparameter differences.
> > >
> > > This is precisely where the significance of our work lies. With the proposed Lookaround method, the weights are continuously averaged throughout the training process, ensuring their alignment within the same loss basin and convergence towards flatter minima.
> > >
> > > >  Moreover, this procedure seems to lose the benefits of pre-training since it shows much worse results than the ones in the literature (84.5% in [2] and 86.4% in [3] v.s. yours 82%)
> > >
> > > Here we would like to, again, remind the reviewer that Lookaround is proposed for a quite different problem setting compared to Model Soups. Model Soups is designed for more efficient utilization of existing models, assuming that these models have already been pre-trained on specific datasets. Lookaround, on the other hand, serves as an optimizer for training deep models, devoid of any assumptions regarding the initialization of the model parameters.
> > >
> > > Regarding the difference between our ResNet50 results on CIFAR100 and the results in [2] and [3], the reason is that both [2] and [3] conducted about 50 grid searches on learning rate and weight decay for the experiment, while we didn't.
> > >
> > > > Also, did you update batch norm parameters for model soups?
> > >
> > > Following the setting in the original Model Soups paper and code, we did not make additional updates to the batch norm parameters.

---

> > > > ### Author Response · Authors · 2023-08-18
> > > > **Reply to the second comment [2/2]**
> > > >
> > > > >  Your results for logit ensembles are quite surprising, hence, I am not fully convinced by the experiments on one dataset. Did you make similar experiments on other datasets? Did you try to compare Lookaround to an ensemble of SWA models?
> > > >
> > > > We conducted a full experimental validation of Logit Ensemble and SWA under a new dataset Stanford Cars, as shown in Table S4. Under the Stanford Cars dataset, Lookaround still performs better than Logit Ensemble. Additionally, adding SWA into the training process of Lookaround can further improve performance. Under the limit of single-model inference, Lookaround+SWA achieves a Top-1 accuracy of 91.04%, exceeding the optimal 88.11% (SWA with AutoAugment).
> > > >
> > > >
> > > > Table S4: Comparison between different training strategies with ResNet18 on Stanford Cars dataset. Here "H" denotes random horizontal flip data augmentation. "R" denotes RandAugment data augmentation. "A" denotes AutoAugment data augmentation. "Logit Ensemble + SWA" denotes a logit ensemble approach that employs SWA models, which have been acquired through various data augmentation techniques.
> > > >
> > > > |       Method       | Data Augmentation Strategy | Top-1 Acc | Top-5 Acc | Training Budget | Inference Budget |
> > > > | :----------------: | :------------------------: | :-------: | :-------: | :-------------: | :--------------: |
> > > > |        SGDM        |            "H"             |   86.77   |   97.52   |       1x        |        1x        |
> > > > |        SWA         |            "H"             |   87.60   |   98.21   |       1x        |        1x        |
> > > > |        SGDM        |            "R"             |   87.02   |   98.01   |       1x        |        1x        |
> > > > |        SWA         |            "R"             |   88.00   |   98.21   |       1x        |        1x        |
> > > > |        SGDM        |            "A"             |   87.63   |   98.05   |       1x        |        1x        |
> > > > |        SWA         |            "A"             |   88.11   |   98.15   |       1x        |        1x        |
> > > > |   Logit Ensemble   |         "H" + "R"          |   89.24   |   98.44   |       2x        |        2x        |
> > > > |     Lookaround     |         "H" + "R"          |   89.35   |   98.47   |       2x        |        1x        |
> > > > |   Logit Ensemble   |      "H" + "R" + "A"       |   90.19   |   98.67   |       3x        |        3x        |
> > > > |     Lookaround     |      "H" + "R" + "A"       |   90.76   |   98.78   |       3x        |        1x        |
> > > > | Logit Ensemble+SWA |      "H" + "R" + "A"       |   90.39   |   98.80   |       3x        |        3x        |
> > > > | **Lookaround+SWA** |    **"H" + "R" + "A"**     | **91.04** | **98.86** |     **3x**      |      **1x**      |
> > > >
> > > > [1] Wortsman M, Ilharco G, Gadre S Y, et al. Model soups: averaging weights of multiple fine-tuned models improves accuracy without increasing inference time.
> > > > [2] Kornblith at. al. Do Better ImageNet Models Transfer Better? CVPR 2019.
> > > > [3] Grill et. al. Bootstrap Your Own Latent A New Approach to Self-Supervised Learning. NeurIPS 2020.

---

> > > > > ### Comment · Reviewer_vFuf · 2023-08-18
> > > > >
> > > > > Taking into account additional experimental results provided during the rebuttal, I've increased my score to reflect my increased confidence in the effectiveness of the proposed method. However, I still have a significant concern regarding the absence of adequate comparison with using the branching and averaging only at the end of training (as was done in model soups, for example).
> > > > >
> > > > > I think such a baseline is important since the proposed Lookaround algorithm is generally based on two ideas: 1) averaging during training helps (as was shown in Lookahead), and 2) averaging several independently trained branches works better than averaging several checkpoints from the same training trajectory (as was shown in model soups and DiWA). The experiments in the paper show that changing the averaging strategy helps (comparison with Lookahead) but do not show that using branching and averaging through the whole training process is better than just using it at the end of training. Ideally, I think three experiments are needed to show that:
> > > > > * Comparison with model soups in the transfer learning setup. I do not find current results convincing because of the low quality of baselines and random quality of uniform model soups. Appropriate training hyperparameters should be chosen for both SGDM and model soups baselines. Now both methods clearly underperform because of the sub-optimal fine-tuning procedure. Also, batch norm statistics (running mean and variance) should always be recomputed after model averaging (see SWA [1] and REPAIR [2]). The authors of the model soups did not do it because they used networks with layer norm, which does not have these statistics. I did some experiments with averaging several fine-tuned ResNet-50 on CIFAR-100 (standard PyTorch supervised pre-training on ImageNet and standard fine-tuning procedure from BYOL): even though uniform model soups do not always improve the results compared to one fine-tuned model, their predictions are much better than random after updating batch norm statistics (~84-84,5% both for individual models and uniform model soups of 5 models, which is higher than the results of Lookaround provided in the paper).
> > > > > * Ablation study where models are trained together at first, and the branching and averaging strategy is used only after some epoch N. If the averaging is only used once at the end of the training, that would be an analog of model soups for a non-transfer learning setup.
> > > > > * Comparison to DART [3]. Lookaround is extremely close to DART, which was published almost three months before the NeurIPS submission deadline. Hence, formally, the authors should provide an experimental comparison of the methods. Given the closeness of the DART release date to the two-month prior mark, I would not see that as a critical issue if the first two experiments were provided in the paper. However, in their absence, I can not be sure that the proposed idea of using branching and averaging through the whole training procedure works better than DART and hence have concerns regarding the novelty of the work.
> > > > >
> > > > > Generally, I still have concerns about the quality of the baselines (both ResNet-50 and VIT baselines in Table 4 significantly underperform, see, e.g., Table 10 in [4]), which I believe is an important issue for the paper proposing a new training technique. Also, I think batch norm statistics recomputation should be taken into account not only in the baselines but in the proposed method itself.
> > > > >
> > > > > The new baselines on ImageNet look much better! Regarding the ensemble experiments, I would recommend broadening this part of the paper and adding even more experiments since the current results look promising.
> > > > >
> > > > > [1] Garipov et.al. Loss Surfaces, Mode Connectivity, and Fast Ensembling of DNNs. NeurIPS 2018.\
> > > > > [2] Jordan et.al. REPAIR: REnormalizing Permuted Activations for Interpolation Repair. ICLR 2023.\
> > > > > [3] Jain et.al. DART: Diversify-Aggregate-Repeat Training Improves Generalization of Neural Networks. CVPR 2023.\
> > > > > [4] Chen et.al. When Vision Transformers Outperform ResNets without Pre-training or Strong Data Augmentations. ICLR 2022.

---

> > > > > > ### Author Response · Authors · 2023-08-20
> > > > > > **Reply to the third comment.**
> > > > > >
> > > > > > Thanks for your constructive comments. We sincerely hope our responses below fully address your questions.
> > > > > >
> > > > > > > Comparison with Model Soups in the transfer learning setup.
> > > > > >
> > > > > > Based on the pre-training settings given in Table-9 of reference paper [1], we re-conducted the Model Soups experiment, and the results are shown in Table S5. Specifically, we selected a more suitable learning rate and a Cosine learning rate scheduler.
> > > > > >
> > > > > > Thank you for introducing the "Update Batch Normalization Parameters" technique. We found it to be highly beneficial for Model Soups when combining multiple models. Furthermore, we conducted Lookaround experiments under new hyperparameters. We discovered that Lookaround with only three augmentation methods surpasses the accuracy of Uniform Soups under standard configurations, but still falls short of Uniform Soups$^+$ and Greedy Soups.
> > > > > >
> > > > > > Table S5: Comparisons between Lookaround, Model Soups on ResNet50. "M=18" represents 18 different hyperparameters. "M=3" represents that Lookaround is trained using 3 different data augmentation techniques. 18 different configurations come from the combinations of three data augmentations (RandomHorizontalFlip, RandAugment, AutoAugment), three initial learning rates: $\{0.01, 0.003, 0.001\}$  and label smooth or not. Unifrom Soups$^+$ represents that we use the top-10 accuracy models for uniform soups (the accuracy of these models ranges from 83.91% to 84.64%).
> > > > > > || Update BN | Top-1 Acc | Top-5 Acc |
> > > > > > | :---------------: | :-------: | :-------: | :-------: |
> > > > > > |Unifrom Soups (M=18)  |&#x2717;     |82.93|   97.44|
> > > > > > |Unifrom Soups (M=18)  |&#x2713;    |84.53|   98.10|
> > > > > > |Unifrom Soups$^+$ (M=18) |&#x2717;|84.60|97.81|
> > > > > > |Unifrom Soups$^+$ (M=18) |&#x2713;|85.57|98.04|
> > > > > > |Greedy Soups (M=18)   |&#x2717;|85.52|98.02|
> > > > > > |Greedy Soups (M=18)   |&#x2713;|86.09|98.11|
> > > > > > | Lookaround (M=3)|-| 85.20|97.15|
> > > > > >
> > > > > >
> > > > > > > Ablation study where models are trained together at first, and the branching and averaging strategy is used only after some epoch N.
> > > > > >
> > > > > > We conducted this experiment using ResNet50 on the CIFAR100 dataset. Unlike DART (where, during the actual training process of the Lookaround optimizer, different branches share the mean and variance of the Batch Normalizaition layer, differing from DART), we observed that the earlier the average step is executed, the better the final model's performance.
> > > > > >
> > > > > > Table S6: The impact of initiating the "average step" at different epochs on the performance of Lookaround with ResNet50 on CIFAR100.
> > > > > > |Start Epoch|0|  40|80|120 | 160|
> > > > > > | :---------:|:---:| :---:| :---:| :---: | :---: |
> > > > > > |Top-1| 81.60 | 80.40 | 80.09 | 78.97 | 77.35 |
> > > > > > |Top-5| 95.99 | 95.53 | 95.30 | 94.87 | 94.30 |
> > > > > >
> > > > > > > Comparison to DART.
> > > > > >
> > > > > > We compared Lookaround with DART with the same data augmentation, the results are shown in Table S7. And our results were better than DART, the reason behind this may come from the fact that we used more frequent "average steps" during training and started "average steps" earlier in the cycle. More frequent "average steps" are more conducive to the training of Lookaround, which is introduced and described in Figure 6 in our paper.
> > > > > >
> > > > > > Table S7: Comparison with the DART method with ResNet18 on CIFAR10 and CIFAR100. The results of DART$^+$ are taken from the original paper.
> > > > > > | | CIFAR10 | CIFAR100 |
> > > > > > |:--------:|:-----:|:------:|
> > > > > > |DART$^+$|97.14|82.89|
> > > > > > |Lookaround |97.22|83.16|
> > > > > >
> > > > > > > Quality concerns regarding the baselines (both ResNet-50 and ViT) in Table 4.
> > > > > >
> > > > > > We apologize that, in the early stages of the experiment, we focused too much on comparing different methods and overlooked the importance of establishing a good baseline for the pre-training process. Therefore, we restarted the pre-training experiment based on the settings in [1] and [2], the results are shown on Table S8.
> > > > > >
> > > > > > In our comparison experiments with the new baseline, we found that the results of ViT-B/16 under different optimizers were better than those reported in the original paper[2]. This is mainly due to our use of more data augmentation during the training process, aiming for a fair comparison with Lookaround. This result also indicates that Lookaround can still achieve better performance under improved baseline conditions.
> > > > > >
> > > > > > Table S8: The test set accuracy under the training procedure with pre-trained models using ImageNet-1k weights. The results of ViT-B/16$^+$ are taken from [2].
> > > > > > | Backbone |   Method   | CIFAR10 | CIFAR100 |
> > > > > > | :------: | :--------: | :-----: | :------: |
> > > > > > |ResNet50|SGDM|97.55|84.50|
> > > > > > |ResNet50|SWA |97.48|84.81|
> > > > > > |ResNet50|Lookahead|97.65  |84.78|
> > > > > > |ResNet50|**Lookaround** |**97.82**|**85.20**|
> > > > > > | ViT-B/16 | Adam |  98.34  |  91.55|
> > > > > > | ViT-B/16$^+$ | Adam| 98.13 | 87.13 |
> > > > > > | ViT-B/16 |SWA|  98.47  |  91.32|
> > > > > > | ViT-B/16 | Lookahead  |98.51|  91.76 |
> > > > > > | ViT-B/16 | **Lookaround** |**98.71**| **92.21**|
> > > > > >
> > > > > > [1] When Vision Transformers Outperform ResNets without Pre-training or Strong Data Augmentations.
> > > > > > [2] An Image is Worth 16x16 Words: Transformers for Image Recognition at Scale.

---

> > > > > > ### Author Response · Authors · 2023-08-21
> > > > > >
> > > > > > Dear reviewer,
> > > > > >
> > > > > > We have conducted new ablation and finetuning experiments and compared our method with both Model Soups and DART. We look forward to your prompt reevaluation and response, as there remains but a scant half-day in the rebuttal stage.
> > > > > >
> > > > > > Thank you once again for your ongoing suggestions and concerns during the rebuttal phase, which will help make Lookaround more reliable.
> > > > > >
> > > > > > Best regards,
> > > > > > The authors of Lookaround

---

> > > > > > > ### Comment · Reviewer_vFuf · 2023-08-21
> > > > > > >
> > > > > > > Now most of my concerns are addressed, hence, I have decided to raise my score further. After the rebuttal and discussions, the paper will need to be seriously revised to make the theoretical part more clear and to include all the new experimental results, therefore, I keep the score at borderline acceptance.

---

### Official Review · Reviewer_vocv · 2023-07-07

**Soundness:** 3 good
**Presentation:** 3 good
**Contribution:** 2 fair
**Rating:** 6
**Confidence:** 3

**Summary:**

This article introduces the Lookaround Optimizer, a novel optimization algorithm for deep neural networks. The Lookaround Optimizer is based on the idea of lookaround, which involves maintaining two sets of weights. The first set of weights is updated using the standard gradient descent algorithm, while the second set of weights is updated using the averaged gradients of the first weight. The Lookaround Optimizer has been shown to improve the generalization performance of deep neural networks, and it outperforms other state-of-the-art optimization algorithms on a variety of benchmark datasets. The authors also provide theoretical analysis of the Lookaround Optimizer. Overall, the Lookaround Optimizer is a promising approach for improving the training of deep neural networks.

**Strengths:**

- Lookaound Optimizer performs better than other state-of-the-art optimization algorithms (Lookahead, SAM) on multiple benchmark datasets.
- The proposed optimizer seems to be model-free and can be applied to various computer vision scenarios.
- The paper is well-written and easy to understand.

**Weaknesses:**

- While the authors have demonstrated fascinating performance on benchmarks, the technical innovation of this paper is limited. Similar ideas that utilize multiple models have already in proposed in varioius area, especially in meta learning.
- With that being said, while the authors have mentioned the connections to these related topics, they did not compare with some of them. For example, I think at least the authors should compare the performance of Lookaround Optimizer with Model Soups (i.e., train multiple copies of models using the same initialization, the same order of data, but with different augmentation), and other model merging techniques like Model Ratatouille.
- The training time will be a bottleneck for applying this method as the authors  acknowledged.
- Another limitation is that, the augmentation seems to be a must for this method. Therefore, it seems that this method is not applicable (at least not straightforward) when we try to apply it on other modality, such as language and speech.

**Questions:**

See above

**Limitations:**

The authors have addressed the limitation.

---

> ### Author Rebuttal · Authors · 2023-08-10
>
> Thank you for your constructive comments and suggestions. In the following, your comments are first stated and then followed by our point-by-point responses.
>
> **Q1:  Similar ideas that utilize multiple models have already in proposed in varioius area, especially in meta learning.**
>
> Thanks for the feedback. Broadly speaking, the goal of meta-learning is to train a model on a diverse range of learning tasks, such that it can quickly solve new learning tasks. To the best of our knowledge, the closest work with the proposed Lookaround is perhaps the MAML-based approaches. However, Lookaround distinguishes itself significantly from MAML in at least the following two aspects:
> * **Problem Settings.** MAML is motivated by learning the common knowledge across tasks, such that new tasks can be quickly solved. In other words, MAML typically assumes the availability of massive tasks for training, with the ultimate goal of solving new tasks with only a few instances. Lookaround, like SGD, Lookhead, or some other optimizers, focuses on addressing general optimization problems.
> * **Methodologies.** MAML trains the initial parameters of the model to maximize its performance on a new task after updating the parameters through one or more gradient steps, utilizing a small amount of data from that particular task. In contrast, Lookaround simultaneously trains multiple models on different augmentations and periodically averages their weights to obtain a final model with enhanced generalization.
>
> **Q2: Compare the performance of Lookaround optimizer with Model Soups and Model Ratatouille.**
>
> We greatly appreciate the reviewer's suggestion. Here we provide the experimental results, details and discussions as follows.
>
> Table S1: Comparisons between Lookaround, Model Soups, and Model Ratatouille on ResNet50. "M=18" represents 18 different hyperparameters. "M=3" represents that Lookaround is trained using 3 different data augmentation techniques.
> |   Optimizer    |        Method (M=18)        | CIFAR100  |
> | :------------: | :-------------------------: | :-------: |
> |      SGDM      |    Model Soups (Greedy)     |   82.07   |
> |      SGDM      |    Model Soups (Uniform)    |   1.00    |
> |     AdamW      |    Model Soups (Greedy)     |   79.18   |
> |     AdamW      |    Model Soups (Uniform)    |   78.28   |
> |     AdamW      | Model Ratatouille (Greedy)  |   80.31   |
> |     AdamW      | Model Ratatouille (Uniform) |   1.58    |
> |   Lookaround |      Lookaround (M=3)       | **83.62** |
>
> **Experimental Details**：18 different configurations come from the combinations of three data augmentations (RandomHorizontalFlip, RandAugment, AutoAugment), three initial learning rates: ($\{0.2, 0.1, 0.05\}$ for SGDM, $\{5e^{-5},2e^{-5},e^{-5}\}$ for AdamW) and label smooth or not. For Model Ratatouille, we choose Stanford cars196, Flowers102, and CIFAR10 as auxiliary domains, and CIFAR100 as the target domain for further fine-tuning.
>
> **Result Discussions**: From Table S1, it can be easily seen that Lookaround *outperforms Model Soups and Model Ratatouille* significantly. Another notable result is that both Model Soups and Model Ratatouille easily break down in uniform averaging strategies (Model Soups with SGDM achieves $1\%$, and Model Ratatouille with AdamW achieves $1.58\%$). In contrast, Lookaround periodically averages the weights during the entire training process, which effectively bypasses the pitfalls of weight averaging between independently trained models.
>
> While Lookaround outperforms Model Soups and Model Ratatouille, it is still necessary to note that *Lookaround is proposed for a quite different problem setting compared to the other two methods*. Model Soups and Model Ratatouille are designed for more efficient utilization of existing models, assuming that these models have already been pre-trained on specific datasets. Lookaround, on the other hand, serves as an optimizer for training deep models, devoid of any assumptions regarding the initialization of the model parameters.
>
> **Q3: The training time will be a bottleneck for applying Lookaround.**
>
> We agree that it is imperative to establish equitable comparisons in computational analysis among different approaches, which is precisely what we've accomplished. Given that the proposed Lookaround utilizes $d$ times the amount of data, we took care that in our primary experiments, all other methods were also trained using $d$ times the data volume. Specifically, within a single epoch, both the proposed Lookaround and the competitors undergo training on an identical $d$ times the data augmentations. With such a setup, we guarantee consistency in the data volume utilized by each method, thereby ensuring fair comparisons in terms of computation.
>
> In the limitation section, we mentioned that Lookaround is limited by the additional cost, considering that Lookaround usually needs more training time to converage than its competitors. However, if restricted to the same training budget, Lookaround still outperforms the others even it have not reached its own optimum. We will provide further clarity on this aspect in the revised paper.
>
> **Q4:  It seems that this method is not applicable on other modality.**
>
> We wholeheartedly agree that exploring the application of Lookaround in other modalities, such as NLP, holds great promise. Indeed, there are existing methods of data augmentation in language tasks, such as EDA[1], TTA[2]. While it may not be a direct and straightforward adaptation, the underlying idea of Lookaround can be explored and applied in innovative ways to these modalities. We eagerly anticipate future advancements and the emergence of new approaches that harness the concept of Lookaround in diverse modalities.
>
> [1] Easy Data Augmentation Techniques for Boosting Performance on Text Classification Tasks, EMNLP-IJCNLP.
> [2] Text AutoAugment: Learning Compositional Augmentation Policy for Text Classification, EMNLP.

---

> ### Author Response · Authors · 2023-08-18
>
> Dear reviewer,
>
> We are glad that the reviewer appreciates our attempt, and sincerely thank the reviewer for the constructive comments.
>
> As suggested, we have elucidated the distinctions between Lookaround and meta-learning approaches and have conducted an exhaustive comparison of Lookaround with other methods such as Model Soups and Model Ratatouille.
>
> Since two-thirds of the allocated time for reviewer-author discussion has already elapsed, we sincerely look forward to your reevaluation of our work and would very appreciate it if you could raise your score to boost our chance of more exposure to the community. Thank you very much!
>
> Best regards,
> The authors of Lookaround

---

> > ### Comment · Reviewer_vocv · 2023-08-19
> > **Acknowledgement**
> >
> > Thanks for the detailed responses! I have looked in to the responses and I think that they have addressed my concerns. However, I have to admit that I might have overlooked some performance gap regarding the baselines. Therefore, I will increase my score and lower my confidence given this controversial situation.

---

### Official Review · Reviewer_HsP8 · 2023-07-12

**Soundness:** 2 fair
**Presentation:** 3 good
**Contribution:** 3 good
**Rating:** 5
**Confidence:** 2

**Summary:**

Flatness-aware optimizers have gained significant attention in the field of research for training deep neural networks that are robust. Weight Averaging (WA) is a popular approach to finding solutions within the flat regions of the loss surface. However, previous WA methods have two limitations. First, when WA is performed within a single optimization trajectory after training convergence, resulting in limited functional diversity among the averaged members. Second, when WA is performed across diverse modes, the averaged weights may negatively impact performance. To address these challenges, this paper introduces a new method called Lookaround Optimizer, which aims to overcome these limitations.

**Strengths:**

Originality

- Proposed optimizer seems to be relatively simple and straightforward but still attractive.
- This article presents a theoretical analysis regarding the variance of the steady state.

Clarity

- The paper is written effectively, ensuring high accessibility for readers.
- Methods are simple and easy to follow.

**Weaknesses:**

Experiments

- Regarding the experiments conducted, the authors acknowledged in the Limitation section that their proposed method incurs additional training costs (I think the proposed optimizer requires nearly $d$ times more computational resources when averaging $d$ models). Therefore, for a fair comparison, it would be necessary for other baseline methods to undergo additional training epochs as well. However, it appears that the authors used the same number of training epochs for all the experiments.

- The overall results of the experiments are rather counterintuitive. Particularly in Table 3, the reported accuracy of the Deep Ensemble method being lower than the accuracy of a single solution trained using the proposed method is quite unexpected. Similarly, in Table 1, the reported performance of flatness-aware optimizers such as SWA [1] and SAM [2] being lower than SGDM contradicts the knowledge and expectations of the research community.

References

[1] Pavel Izmailov, D. A. Podoprikhin, Timur Garipov, Dmitry Vetrov, and Andrew Gordon Wilson. Averaging weights leads to wider optima and better generalization. Conference on Uncertainty in Artificial Intelligence, 2018.

[2] Pierre Foret, Ariel Kleiner, Hossein Mobahi, and Behnam Neyshabur. Sharpness-aware minimization for efficiently improving generalization. International Conference on Learning Representations, 2021.

**Questions:**

Experiments

- The response regarding concerns addressed in the Weakness section will be critical to the final decision.

**Limitations:**

Limitations are adequately addressed in the Limitation section.

---

> ### Author Rebuttal · Authors · 2023-08-10
>
> We sincerely appreciate your comments on our work. We hope the following response will address your concerns.
>
> **Q1: Lookaround uses $d$ times more computation. How does Lookaround fairly compare to other methods?**
>
> We agree that it is imperative to establish equitable comparisons in computational analysis among different approaches, which is precisely what we've accomplished. Given that the proposed Lookaround utilizes $d$ times the amount of data, we took care that in our primary experiments (corresponding to Table 1 to 4 in the paper), all other methods were also trained using $d$ times the data volume. Specifically, within a single epoch, both the proposed Lookaround and the competitors undergo training on an identical $d$ times the data augmentations. With such a setup, we guarantee consistency in the data volume utilized by each method, thereby ensuring fair comparisons in terms of computation. With the same amount of computation, Lookaround outperforms the competitors by a considerable margin, which demonstrates the superiority of Lookaround in efficiency.
>
> In the limitation section, we mentioned that Lookaround is limited by the additional cost, considering that Lookaround usually needs more training time to converage than its competitors. However, if restricted to the same training buget, Lookaround still outperforms the others even it have not reached its own optimum. We will provide further clarity on this aspect in the revised paper.
>
> **Q2: In Table 3, the reported accuracy of the ensemble method being lower than the accuracy of a single solution trained using the proposed method is quite unexpected.**
>
> Thanks for the comment. The proposed Lookaround, as previously mentioned, achieves remarkably superior performance with a single solution to the preceding ensemble method. In the case of Logit Ensemble, results are obtained by ensembling from *independently* trained models. The performance improvement through ensembling is solely derived from the diversity. On the contrary, Lookaround can be perceived as an ensemble (Recall that weight average approximates straightforward ensemble as proven in SWA) of *simultaneously and interdependently* trained models. During the training process, these models benefit from one another by weight averaging and converge towards flatter minima. The performance gain arises not only from the diversity (each model is trained on different augmentations), but also from the superiorioty of the base models. Hence, it is entirely rational that the proposed Lookaround surpasses simple ensembling methods, which should be considered an advantage rather than a disadvantage of our work.
>
> **Q3: Similarly, in Table 1, the reported performance of flatness-aware optimizers such as SWA and SAM being lower than SGDM contradicts the knowledge and expectations of the research community.**
>
> Thanks for raising these concerns. Here we would like to make the following clarifications.
> - For the optimizer SWA, we would like to remind the reviewer that it outperforms SGDM in nearly all our experiments, as evidenced in Table 1, Table 2, and Table 4 in our paper. The only one exception arises in the case of the model VGG19 on CIFAR100, where SWA achieves slightly lower accurcy. These results, however, align quite consistently with the prevailing consensus within the research community.
>
> - For the optimizer SAM, we have carefully checked the experimental configurations and repeated the experiments three times. The results remain consistent across our repeated trials. We speculate that the heavy data augmentation employed in our experiments might be the predominant factor in the inferior performance of SAM. To ensure fair comparisons in computation, both Lookaround and the competitors, including SAM, are evaluated with heavy data augmentations (random horizontal flip+random vertical flip+RandAugment). The heavy data augmentations may serve as a potential distraction for SAM, as both techniques are proposed to enhance generalization. To verify this hypothesis, we conducted a sanity check of our code with only a single type of data augmentaion. The results are provided in Table S1. It can be seen that *SAM significantly ourperforms SGD with only one type of augmentation, which is still consistent with consensus of the research community*.
>
>     Table S1: Comparison of different optimization methods for ResNet18 on various datasets. "H" denotes random horizontal flip data augmentation. "R" denotes RandAugment data augmentation.
>
>     |                  |  SGDM (H) |  SGDM \(R\) |  SAM (H)  |  SAM \(R\)  | Lookaround (R+H) | Lookaround+SAM (R+H)|
>     | :--------------: | :---: | :---: | :---: | :---: | :--------: | :------------: |
>     |     CIFAR100     | 75.72 | 76.14 | 76.84 | 77.08 |   77.20    |     77.55      |
>     |    Flowers102    | 95.93 | 96.54 | 97.01 | 97.35 |   97.65    |     97.75      |
>     | Stanford cars196 | 86.77 | 87.02 | 87.53 | 87.82 |   89.35    |     89.67      |
>
>
> Furthermore, the results listed in Table 1 and Table S1 implies a potential merit of the proposed Lookaround: it is orthogonal to SAM and heavy data augmentation, and can work compatibly with both, as validated by *ours* (Lookaround+heavy augmentation) in Table 1, and *Lookaround+SAM* in Table S1. We hope our response can successfully address your concerns and strengthen your confidence in our work.

---

> > ### Author Response · Authors · 2023-08-15
> >
> > Dear reviewer,
> >
> > We extend our sincerest gratitude for the questions you raised during the initial rebuttal stage. We have taken great care in providing thorough answers to address your concerns. As we approach the halfway point of the author-reviewer discussion phase, we are eager to receive further feedback and engage in fruitful discussions with you.
> >
> > Best regards,
> > The authors of Lookaround

---

> > > ### Comment · Reviewer_HsP8 · 2023-08-16
> > >
> > > I appreciate the thorough clarification and the inclusion of extra experiments. It appears that my concerns align with those expressed by reviewer vFuf and d83u. I will await the outcome of the discussion between the authors and vFuf, d83u before making my final evaluation.

---

> > > > ### Author Response · Authors · 2023-08-18
> > > > **Thanks for your reply!**
> > > >
> > > > We sincerely appreciate your feedback on our rebuttal. To address your concerns and those of other reviewers regarding Logit Ensemble and SWA, we have conducted more comprehensive experiments and provided clarifications. Through our experimental results, we have demonstrated the superiority of the Lookaround method over Logit Ensemble and SWA across different datasets. Moreover, we have further explained that our method can be combined with Logit Ensemble and SWA for even better performance. We sincerely hope you will carefully review the materials we have provided and re-evaluate our work.
> > > >
> > > > We understand that you may take into account the perspectives of others when making your assessment, which is perfectly reasonable. However, we also hope that you will make your judgment based on the new empirical data and analysis that we present, drawing on your expertise. We believe this will contribute to a fairer and more accurate review process. If you have any further questions or require additional clarification, please don't hesitate to contact us. We would be very grateful if you could provide a more impartial and objective evaluation of our work.
> > > >
> > > > Thank you again for your time and effort. We look forward to your response!

---

> > > > ### Author Response · Authors · 2023-08-21
> > > >
> > > > Dear reviewer,
> > > >
> > > > We have addressed the concerns raised by other reviewers about our paper and have received their positive responses. We are looking forward to your reevaluation of our manuscript, drawing upon your expertise, as there remains but a scant half-day in the rebuttal stage. We extend our gratitude to you once more.
> > > >
> > > > Best regards,
> > > > The authors of Lookaround

---

> > > > > ### Comment · Reviewer_HsP8 · 2023-08-21
> > > > >
> > > > > I would like to express my gratitude to the authors for their dedication in conducting supplementary experiments during the review process. Due to their commendable efforts, I am inclined to raise my rating to borderline acceptance. However, considering the necessity for augmentation techniques and the somewhat inconclusive nature of the experimental outcomes, my confidence level slightly diminished.

---

### Decision · Program_Chairs · 2023-09-21

**Decision:**

Accept (poster)

**Comment:**

This paper proposes a new  Lookaround optimizer for network training. It is inspired by weight averaging (WA) techniques in deep learning, and thus it looks around nearby points by performing multiple gradient computations and then averages them to obtain better generalizing solutions. Experimental results show the effectiveness of Lookaround. All the reviewers unanimously accepted the paper, and so did the final decision. The authors should also revise their paper according to the reviewers' suggestions in the final version.